# Balance between breadth and depth in human many-alternative decisions

**Alice Vidal[1,2]\*, Salvador Soto-Faraco[1,3], Rubén Moreno-Bote[1,4]**

[1]Center for Brain and Cognition, and Department of Information and Communication Technologies, Universitat Pompeu Fabra, Barcelona, Spain; [2]Department of Experimental and Health Sciences, Universitat Pompeu Fabra, Barcelona, Spain; [3]Insitució Catalana de la Recerca i Estudis Avançats (ICREA), Barcelona, Spain; [4]Serra Húnter Fellow Programme, Universitat Pompeu Fabra, Barcelona, Spain

**Abstract** Many everyday life decisions require allocating finite resources, such as attention or time, to examine multiple available options, like choosing a food supplier online. In cases like these, resources can be spread across many options (breadth) or focused on a few of them (depth). Whilst theoretical work has described how finite resources should be allocated to maximize utility in these problems, evidence about how humans balance breadth and depth is currently lacking. We introduce a novel experimental paradigm where humans make a many-alternative decision under finite resources. In an imaginary scenario, participants allocate a finite budget to sample amongst multiple apricot suppliers in order to estimate the quality of their fruits, and ultimately choose the best one. We found that at low budget capacity participants sample as many suppliers as possible, and thus prefer breadth, whereas at high capacities participants sample just a few chosen alternatives in depth, and intentionally ignore the rest. The number of alternatives sampled increases with capacity following a power law with an exponent close to 3/4. In richer environments, where good outcomes are more likely, humans further favour depth. Participants deviate from optimality and tend to allocate capacity amongst the selected alternatives more homogeneously than it would be optimal, but the impact on the outcome is small. Overall, our results undercover a rich phenomenology of close-to-optimal behaviour and biases in complex choices.

**\*For correspondence:**
alice.vidal@upf.edu

**Competing interest:** The authors declare that no competing interests exist.

## Editor's evaluation

The authors describe human behavior in a novel task to understand how humans seek information about uncertain options when having a limited sampling budget – the 'breadth-depth' trade-off. They show that human information search approximates the optimal allocation strategy, but deviates from it by favoring breadth in poor environments and depth in rich environments. This study will likely be of interest to a broad range of behavioral and cognitive neuroscientists.

## Introduction

When choosing an online food supplier as we settle into a new area, we need to trade off the number of alternative shops that we check with the time or money that we want to invest in each of them to learn about the quality of their products. Distributing resources widely (breadth search) allows us to sample many suppliers but very superficially, thus limiting our ability to distinguish which one is best. Allocating our resources to check just a few suppliers (depth search) allow us to learn detailed information but only from a few, at the risk of neglecting potentially much better ones. Striking the right balance between breadth and depth is critical in countless other endeavours such as when selecting which courses to register in college (*Schwartz et al., 2009*) or developing marketing strategies

(**Turner et al., 1955**). Its implications are far reaching when it comes to understand behaviours on the internet, where breadth and depth search has been related to navigation through lists of search results (**Klöckner et al., 2004**) or web site menus (**Miller, 1991**).

Despite its relevance to understand how humans make decisions under finite resources, it is remarkable that the *breadth-depth* (BD) dilemma has mostly been investigated outside cognitive neuroscience (**Moreno-Bote et al., 2020**), in contrast to other well-studied trade-offs like speed-accuracy and exploration-exploitation (**Cohen et al., 2007**; **Costa et al., 2019**; **Daw et al., 2006**; **Ebitz et al., 2018**; **Wilson et al., 2014**). The BD dilemma underlies virtually all cognitive problems, from allocating attention amongst multiple alternatives in multi-choice decision making (**Busemeyer et al., 2019**; **Hick, 1958**; **Proctor and Schneider, 2018**), splitting encoding precision to items in working memory (**Joseph et al., 2016**; **Ma et al., 2014**), to dividing cognitive effort into several ongoing subtasks (**Feng et al., 2014**; **Musslick and Cohen, 2021**; **Shenhav et al., 2013**). In all these problems, finite resources, such as attention, memory precision, or amount of control, need to be allocated amongst many potential options simultaneously, making the efficient balancing between breadth and depth a fundamental computational conundrum.

The dynamics of resource allocation can be very complex, and thus it has been studied in decision making in simplified cases with few alternatives (**Callaway et al., 2021b**; **Jang et al., 2021**; **Krajbich et al., 2010**) or in multi-tasking using a low number of simultaneously active tasks (**Musslick and Cohen, 2021**; **Sigman and Dehaene, 2005**). In these and other cases, resource allocation can be changed on the fly if feedback is immediate or is available within very short delays. In some real-life situations, however, feedback about the quality of the allocation is necessarily delayed. This happens for instance in problems such as investing (**Blanchet-Scalliet et al., 2008**; **Reilly et al., 2016**), choosing college (**Schwartz et al., 2009**) or, in ant colonies, sending scouts for exploration (**Pratt et al., 2002**); feedback can come after seconds, days, or even years. As resources should then be allocated beforehand, the dynamic aspect of the allocation is less relevant. One-shot resource allocation is important in cognition as well, as building-in stable attentional or control strategies that work in a plethora of situations could relieve the burden of solving a taxing BD dilemma for optimal resource allocation every time (**Mastrogiuseppe and Moreno-Bote, 2022**).

Previous theoretical work has shown how to optimally trade off breadth and depth over multi-alternative problems in the situations described above, where resources are allocated all at once before feedback is received (**Mastrogiuseppe and Moreno-Bote, 2022**; **Moreno-Bote et al., 2020**; **Ramírez-Ruiz and Moreno-Bote, 2022**). A central result is that the optimal trade-off depends on the search capacity of the agent: while at low capacity resources should be split in as many alternatives as capacity permits (breadth), at high capacity resources should be focused on a relatively small number of selected alternatives so that available resources are more focused (depth) (**Moreno-Bote et al., 2020**). In rich environments, where finding options rendering good outcomes is more likely, depth should be further favoured. Despite of the existence of precise predictions describing ideal behaviours in these scenarios, how humans solve BD trade-offs in many-alternative decision making is largely unknown (**Brown et al., 2008**; **Callaway et al., 2021a**; **Chau et al., 2014**; **Cohen et al., 2017**; **Hawkins et al., 2012**; **Moreno-Bote et al., 2020**; **Roe et al., 2001**; **Usher and McClelland, 2004**; **Vul et al., 2014**).

To fill this gap, we designed a novel, many-alternative task where search capacity was parametrically controlled on a trial-by-trial basis. Human participants are immersed in a context where they are asked to allocate finite search capacity over a large number (several dozens) of apricot suppliers with the goal to choose the best one. We compare human sampling behaviour to optimality (**Moreno-Bote et al., 2020**, see Materials and methods for more details) by using two different ranges of capacities to zoom in relevant regimes. Concerning the adaptation of the sampling strategy as a function of capacity, we expected a pure-breadth behaviour at low capacity followed by a sharp transition towards a trade-off between breadth and depth, characterized by an increase of the number of alternatives sampled with the square root of capacity. We also addressed the effect of environment richness (the overall probability of alternatives rendering good outcomes) on the sampling strategy. We used three different environments with either a majority of poor, neutral, or rich alternatives and expected sampling strategy to shift towards depth as the environment became richer. Finally, we looked at whether systematic deviations from optimality can be observed regarding how samples are

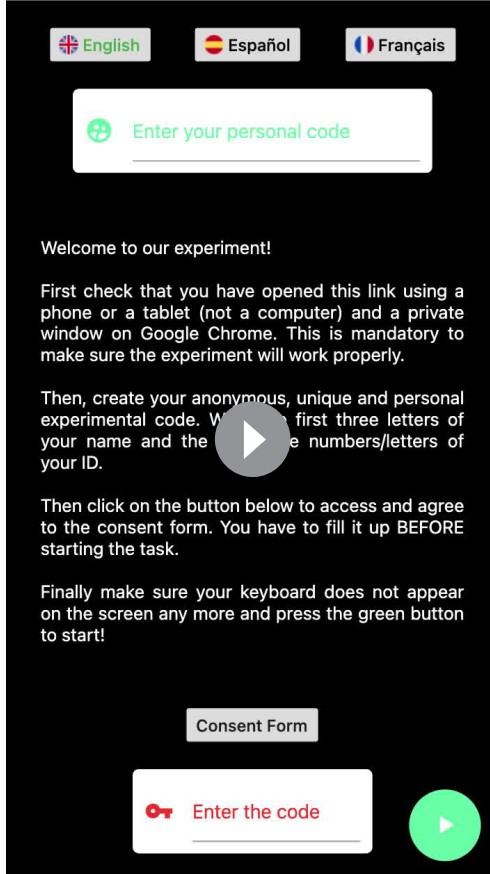

**Video 1.** Demonstration video of the task (design W10).
https://elifesciences.org/articles/76985/figures#video1

distributed amongst each sampled alternative. We employed model comparison to adjudicate between optimal and other heuristic models of behaviour.

## Results
### Humans switch from breadth to a BD balance as capacity increases

We developed a novel experimental approach, the 'BD apricot task', to study many-alternative decisions under uncertainty and limited resources by confronting participants with a BD dilemma. Participants played a gamified version of the task in which they were presented with several virtual apricot suppliers (10 or 32) having different, and unknown, probabilities of good-quality apricots (Materials and methods; see *Video 1*). These probabilities were independently generated for each supplier in each trial from a beta prior distribution. At the beginning of each trial, participants received a budget of a few coins, which defined their sampling capacity in that trial. Each coin was used to buy one apricot from a selected supplier (*Figure 1a*). Once all coins had been spent (*Figure 1b*), participants discovered which of the sampled apricots were of good quality, and which ones were bad (*Figure 1c*), whose outcomes followed a Bernoulli process with the unknown probabilities of good-quality apricots for each supplier. These probabilities were sampled independently for each supplier and trial from a beta distribution. Based on the observed outcomes, participants could estimate the probabilities from

the sampled suppliers, and make a final purchase of 100 apricots from the one they considered to be the best (*Figure 1d*). Only one of the sampled alternatives could be selected for the final purchase. Their goal was to maximize the number of good-quality apricots collected throughout the experiment through the implementation of an informative sampling strategy, adapted both to the sampling capacity and to the environment richness. The task was intuitive and easily grasped by most participants. No instructions about the underlying probability generative model was provided as the context was informative enough to aid task understanding (*Schustek et al., 2019*).

Two different ranges of sampling capacity, narrow (2–10 samples per trial) and wide (2–32), were tested to zoom-in relevant behavioural regimes (see Materials and methods and *Table 1*). For each range, we used three environments differing in the parameters of the beta distribution used to generate probabilities of good-quality apricots for each supplier. By increasing the average probability of good alternatives, we can increase the richness of the environment to move from a 'poor', to a 'neutral', up to a 'rich' environment (beta prior means: poor, 0.25; neutral, 0.50; and rich, 0.75). While environments and capacity ranges were run in a block fashion, capacity in each was chosen randomly in each trial from the underlying range in that block. Environments were tested within and between subjects in two different sets of experiments, to address potential learning effects.

We observed in all environments that participants' sampling behaviour of suppliers follows a pure-breadth strategy at low capacity (*Figure 2A–B*), whereby they sample close to as many suppliers as coins they have. Indeed, at low capacity ($C \sim 2, 3$) the number $M$ of sampled suppliers averaged over trials and participants is close to $C$ (*Figure 2A*) and the ratio $M/C$ reaches very close to 1 (*Figure 2B*). At higher capacities, sampling progressively evolves towards a trade-off between breadth and depth.

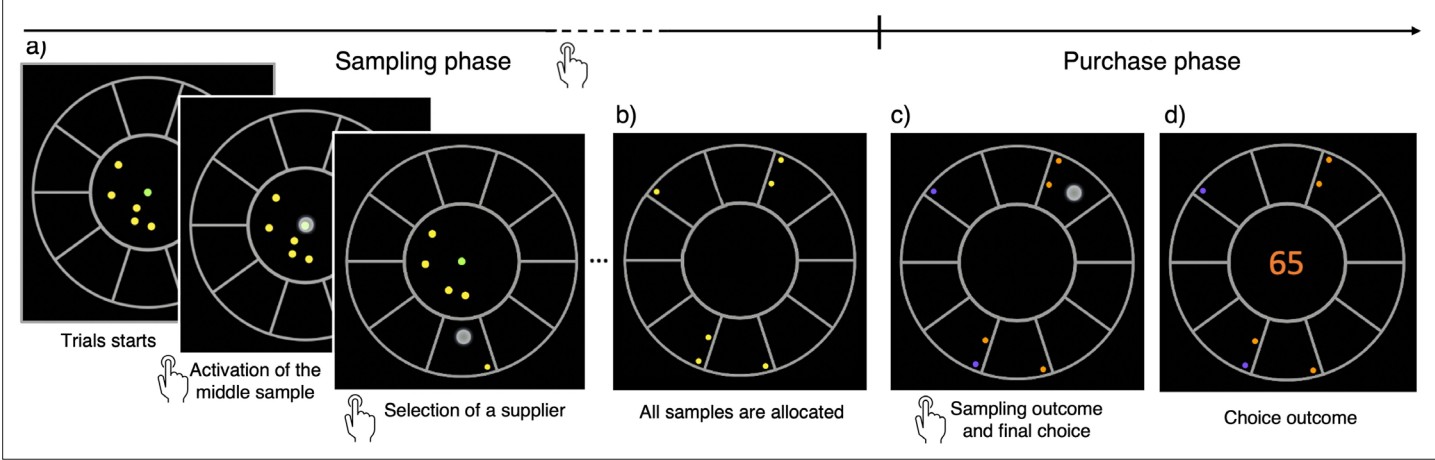

**Figure 1.** The breadth-depth (BD) apricot task. Human participants allocate a finite search capacity (coins) to learn about the quality of good apricots in different suppliers (sampling phase) and then make a final purchase of 100 apricots from one of the sampled suppliers (purchase phase). Each black section of the wheel represents a different supplier. The number of coins represents the search capacity of the participants on each trial and varies randomly from trial to trial within a finite range (see Materials and methods). The available coins (panel a; yellow green dots) at any time during the trial are displayed within the centre of the wheel. To allocate the coins to suppliers, participants have first to click on the designated active coin displayed at the centre (green dot) and then select the supplier to sample from (panel a) – both touch screen events are indicated by a large grey dot. One of the inactive (yellow) coins is then automatically activated and displayed, in green, at the centre. This sequence repeats until all coins are allocated. Then, each of the allocated samples turn either orange, representing a good-quality apricot, or purple, representing a bad-quality apricot (panel c). Finally, after this information is revealed, the participant selects one of the sampled suppliers for the final purchase of 100 apricots (with a touch screen, indicated by a large grey dot) and the choice outcome is immediately displayed (panel d).

Therefore, although participants could have sampled more suppliers to learn about, they preferred to focus sampling capacity on a rather small fraction of suppliers, as shown by the decline of the ratio *M/C* towards values of around 0.4 at the highest capacities tested (*Figure 2B*; right panels). Although these are predicted features and thus signs of optimal behaviour, we did not observe a fast transition between the low- and high-capacity regimes, as previously predicted (*Moreno-Bote et al., 2020*, see also *Appendix 1—figure 1A*). This could simply result from averaging over participants having different transition points, or by having allocation noise, which would smooth out fast transitions. As shown below, models with allocation noise account for the smoothness of this transitions and are better models in predicting the behaviour than the noise-free optimal model.

## Richer environments promote depth

The effect of environmental richness was also clearly visible in our data, with richer environments typically causing a stronger preference for depth sampling (*Figure 2A–B*; different colours), consistent with predictions. Therefore, as it becomes easier for the participant to find good options, they prefer to neglect an even larger number of suppliers and focus capacity to examine fewer ones. We also observed systematic deviations from optimality (*Figure 2A*; black lines). In particular, despite participants' strategy was overall strongly dependent on environment richness, as predicted, this was mostly due to a strong effect of the poor environment (*Figure 2A–B*; red lines), whereas they were not sensitive to the difference between neutral and rich environments (green and blue lines).

**Table 1.** Summary of the experimental designs.

|  | Design | Capacity | N suppliers | N trials | N subject |
|---|---|---|---|---|---|
| W10 | Within-subject | 2–10 | 10 | 216 | 18 |
| B10 | Between-subject | 2–10 | 10 | 72 | 45 |
| W32 | Within-subject | 2,4,8,16,32 | 32 | 120 | 18 |
| B32 | Between-subject | 2,4,8,16,32 | 32 | 40 | 45 |

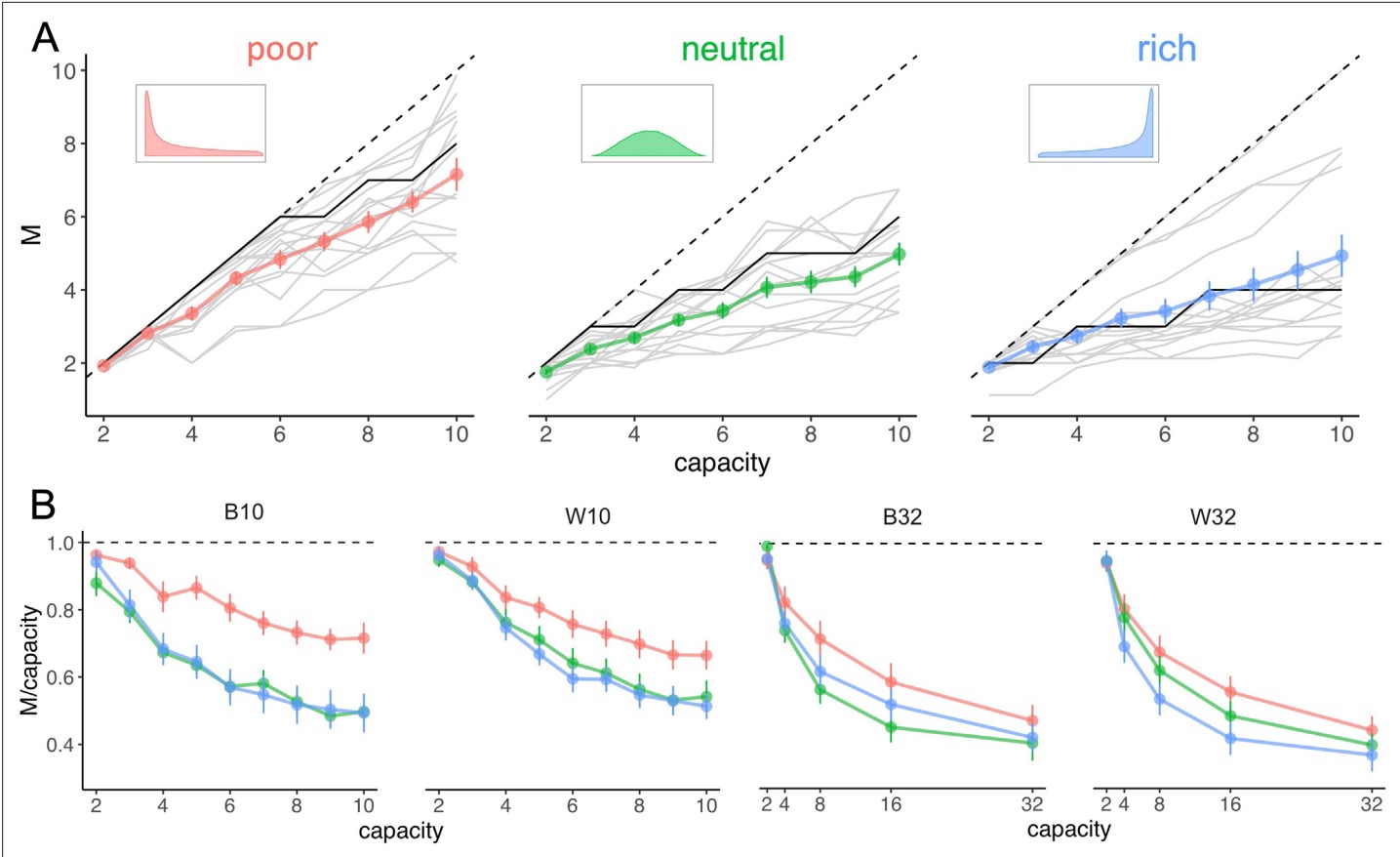

**Figure 2.** Number of sampled suppliers increases with capacity and is strongly sensitive to the richness of the environment, consistent with theoretical predictions. (**A**) Number of alternatives sampled $M$ as a function of capacity averaged across participants (points), for each of the three different environments (colours), for the low-capacity between-subject design (B10). Dashed lines indicate unit slope line. Optimal observer predictions are displayed in black and grey lines represent individual data. Error bars correspond to s.e.m. (**B**) Number of alternatives sampled $M$ divided by capacity as a function of capacity. Colour code as in panel **A**. Samples sizes per environment condition: designs B10 and B32: n=15, designs W10 and W32: n=18.

To test quantitatively the effect of environmental richness on the BD trade-offs, we fitted participant's individual data in each environment using three models. These models were chosen based both on our predictions and on previous observations. First, a piece-wise power-law model (*W*), having a fast transition at some arbitrary capacity value, was selected as it is the one anticipated by the ideal observer. Second, as previously said, we visually noticed that for a majority of participants the transition between pure-breadth and BD trade-off was gradual (see *Figure 2*), so we decided to capture their sampling strategy using two simpler models: a linear model (*L*) and a power-law model (*P*). Indeed, it has been predicted that once the BD trade-off established, the number of alternatives sampled $M$ approximately increases with a power-law behaviour (*Moreno-Bote et al., 2020*). We observed that the power-law model ($R^2_{adj} = 0.96 \pm 0.04$ mean ± s.d.) showed significantly better fits than the linear model ($R^2_{adj} = 0.92 \pm 0.06$ mean ± s.d.; $V = 404$, $p < 2.2 \times 10^{-16}$). Moreover, the linear piece-wise model was not significantly better than the power-law model for all participants in all environments individually (ANOVAs, with $\alpha = .05$).

We compared the exponent estimated from the power-law fits in each environment and experimental design (*Figure 3*) using within- (W10 and 32) or between-subjects (B10 and 32) designs. The results demonstrated a significant effect of the environment on the exponent in all designs (one-way ANOVAs, B10: $F_1 = 13.17$, $p = 7.51 \times 10^{-4}$ ; W10: $F_1 = 31.05$, $p = 3.37 \times 10^{-5}$ ; W32: $F_1 = 15.12$, $p = .0012$), except for the between-subjects design and the larger range of capacities tested (B32: $F_1 = 0.82$, $p = .37$). Overall, effect sizes were larger in designs with narrow (B10: $\omega^2_G = .213$, W10: $\omega^2_G = .121$) compared to wide ranges of capacities (B32: $\omega^2_G = -.004$, W32: $\omega^2_G = .072$) and the non-result observed may stem from an insufficient sample size (achieved power in B32 is 14.7%, against

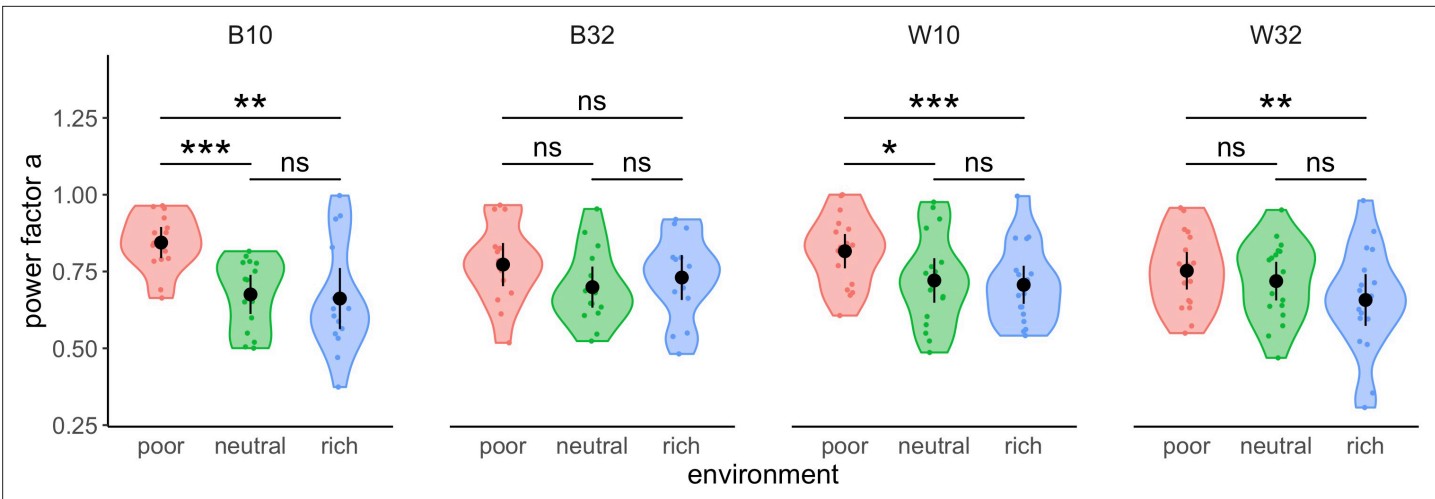

**Figure 3.** Participants' strategy is modulated by the richness of the environment. Distribution of power factors extracted from fitting a linear model to values $M$ vs. capacity in a log-log scale. Colour dots represent subjects, black dots represent means across participants, and bars are 95% confidence intervals. Results of post hoc comparisons are displayed according to adjusted p-values ('ns': p>0.05, '**': p<0.01, '***': p<0.001). Samples sizes per environment condition: designs B10 and B32: n=15, designs W10 and W32: n=18.

97.8% in W32, 95.2% in B10 and 94.7% in W10). As suggested by visual inspection above, there is no significant difference in the exponents between the neutral (median values, W10: $m_{neutral} = 0.72$, B10: $m_{neutral} = 0.68$, W32: $m_{neutral} = 0.70$) and rich (W10: $m_{rich} = 0.71$, B10: $m_{rich} = 0.66$, W32: $m_{rich} = 0.66$) environments in any of the designs (t-tests with Bonferroni correction, W10: $t_{17} = 0.57$, $p_{adj} = 1$, B10: $t_{23.9} = 0.24$, $p_{adj} = 1$, W32: $t_{17} = 2.04$, $p_{adj} = .17$), although strong and significant differences were typically observed between rich and poor (W10: $m_{poor} = 0.82$, B10: $m_{poor} = 0.84$, W32: $m_{poor} = 0.75$) environments (W10: $t_{17} = 5.57$, $p_{adj} = 1.01 \times 10^{-4}$, B10: $t_{20.8} = 3.51$, $p_{adj} = .006$, W32: $t_{17} = 3.89$, $p_{adj} = .004$). Significant differences were also observed between poor and neutral environments in the designs with smaller capacities (W10: $t_{17} = 3.25$, $p_{adj} = .014$, B10: $t_{26.7} = 4.45$, $p_{adj} = 4.11 \times 10^{-4}$). Overall, we observed in the three designs mentioned a decrease of the exponent as the environment gets richer. This suggests an adaptation of participants' sampling behaviour to the environment, with sampling depth increasing with the richness of environment.

## Deviations from optimality

We have demonstrated that, as expected, participants' sampling strategy is modulated by the environment richness. We now investigate whether the strategy used coincides with the optimal sampling behaviour or if some deviations are observable. Pooling the data of the four experimental designs together, we confirmed our previous results and found a significant effect overall of the environment on the exponent of the power law (ANOVA, $F_1 = 21.03$, $p = 8.23 \times 10^{-6}$). Importantly, there

**Table 2.** Participants' sampling strategy significantly deviates from optimality.
Values of the factor $a$ predicted (first row) or observed (averaged across participants ± s.d., second row) depending on capacity using a power-law function with free exponent and fixed intercept (see Materials and methods for more details). Results of comparisons between factors $a$ using one-sample t-tests with Bonferroni corrections (third row) show that participants' sampling strategy extracted from fitting the number of alternatives sampled is significantly tilted towards depth in the poor and neutral (tendency) environments compared to optimality, while in the rich environment participants are sampling in a breather way than predicted.

| Environment Value of power factor $a$ | Poor | Neutral | Rich |
|---|---|---|---|
| Optimal (predicted) | 0.877 | 0.732 | 0.612 |
| Data (observed) | 0.795±0.118 | 0.705±0.127 | 0.688±0.152 |
| Comparison | $t_{65} = -5.66$, $p_{adj} = 3.72 \times 10^{-7}$ | $t_{65} = -1.74$, $p_{adj} = .087$ | $t_{65} = 4.06$, $p_{adj} = 1.36 \times 10^{-4}$ |

was no significant effect of the experimental design (ANOVA, $F_3 = 0.76$, $p = .52$) nor a significant interaction between the environment and the experimental design (ANOVA, $F_3 = 1.40$, $p = .24$), thus confirming that the effect of the environment can be studied on the whole dataset independently of the experimental design. In order to quantify deviations of participants' sampling strategies from the optimal strategy, we fitted the optimal values of the number $M$ of sampled suppliers for all capacities ($C = \{2 - 10, 16, 32\}$) using the power-law model previously described (see Materials and methods) and extracted the power-law exponent. Comparing the observed values of the exponent to the optimal ones (see *Table 2*), we observed that participants' sampling strategy significantly shifted towards excessive depth in the poor environment (t-test, $t_{65} = -5.66$, $p_{adj} = 3.72 \times 10^{-7}$), and a similar tendency occurred in the neutral environment (t-test, $t_{65} = -1.74$, $p_{adj} = .087$). In contrast, participants' sampling strategy deviated significantly from optimality in the direction of excessive breadth in the rich environment (t-test, $t_{65} = 4.06$, $p_{adj} = 1.36 \times 10^{-4}$).

Looking in greater detail at which capacities these deviations occur, we computed the differences between the optimal number $M_{opt}$ of sampled suppliers and the observed $M$ for each capacity and environment and observed again a significant effect of the environment on the differences (Scheirer-Ray-Hare test, $H_2 = 159.67$, $p < 2.2 \times 10^{-16}$), but also a significant effect of capacity ($H_{10} = 98.07$, $p = 2.2 \times 10^{-16}$) and a significant interaction between the environment and capacity ($H_{10} = 84.08$, $p = 7.89 \times 10^{-10}$), showing that the participants' bias towards excessive depth or breadth varies with the environment and the capacity. In particular, we observed that at low capacity, the difference between $M_{opt}$ and the observed $M$ tended to be positive, which is especially visible in the poor and neutral environments (*Appendix 1—figure 1* and exhaustive analyses presented in *Supplementary file 1*). In contrast, at high capacity this difference tends to be negative, which is especially noticeable in the neutral and rich environments.

The results of these analyses indicate some deviations from optimality. Participants have a bias to sample more deeply than optimal at low capacities while this bias is reversed at high capacities. These overall biases could be accounted for by assuming that participants have a biased model of the richness of the environment, not estimating as much as they should the extreme nature of poor or rich environments (*Schustek et al., 2019*).

We next studied whether the deviations from optimality weakened over time or were persistent. We fitted the power-law model to individual BD trade-off in each environment (block) for the first and second halves of each block separately (median split, *Figure 4*, BD trade-offs are presented in

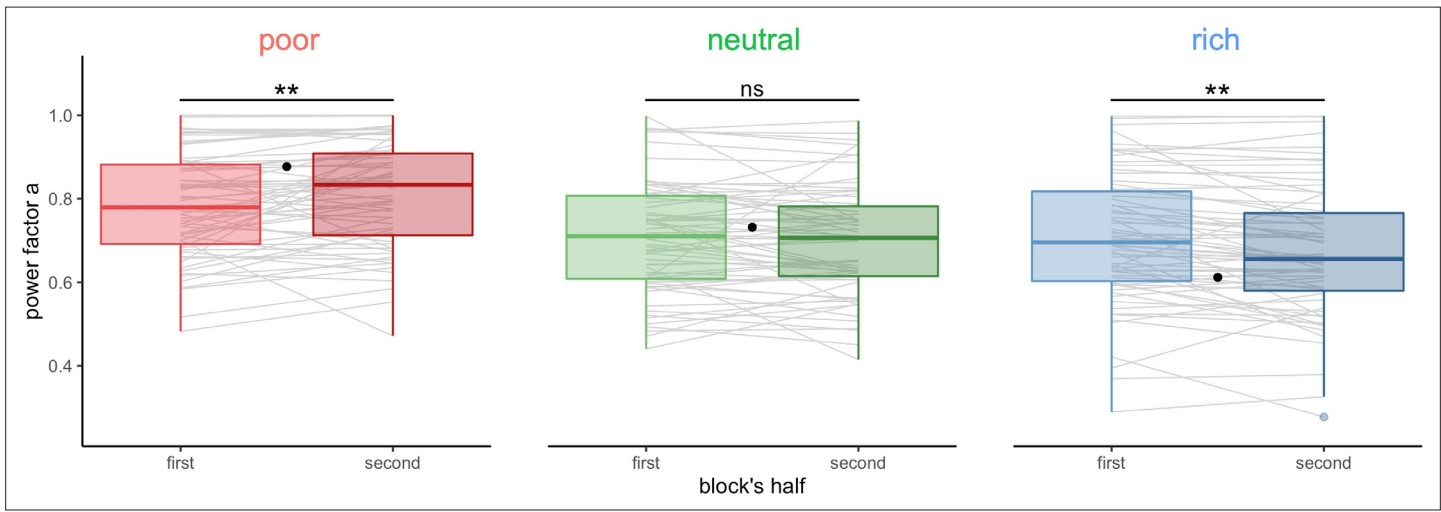

**Figure 4.** Participants' sampling strategy gets closer to the optimal breadth-depth (BD) trade-offs with experience. Distribution of the power factor *a* in the power-law model when fitting the number of alternatives sampled *M* as a function of the capacity in each environment, separately for each block's half (median split on the number of trials). Each line connects a subject, black dots represent the power factor *a* when fitting the optimal BD trade-offs. Results of post hoc comparisons are displayed according to adjusted p-values ('ns': $p_{adj}$ >0.1, '**': $p_{adj}$ <0.01). Lower and upper hinges correspond to the 1st and 3rd quartiles and vertical lines represent the interquartile range (IQR) multipled by 1.5. Sample sizes n=66.

The online version of this article includes the following figure supplement(s) for figure 4:

**Figure supplement 1.** Participants' sampling strategy gets closer to the optimal breadth-depth (BD) trade-offs as time passes within a block.

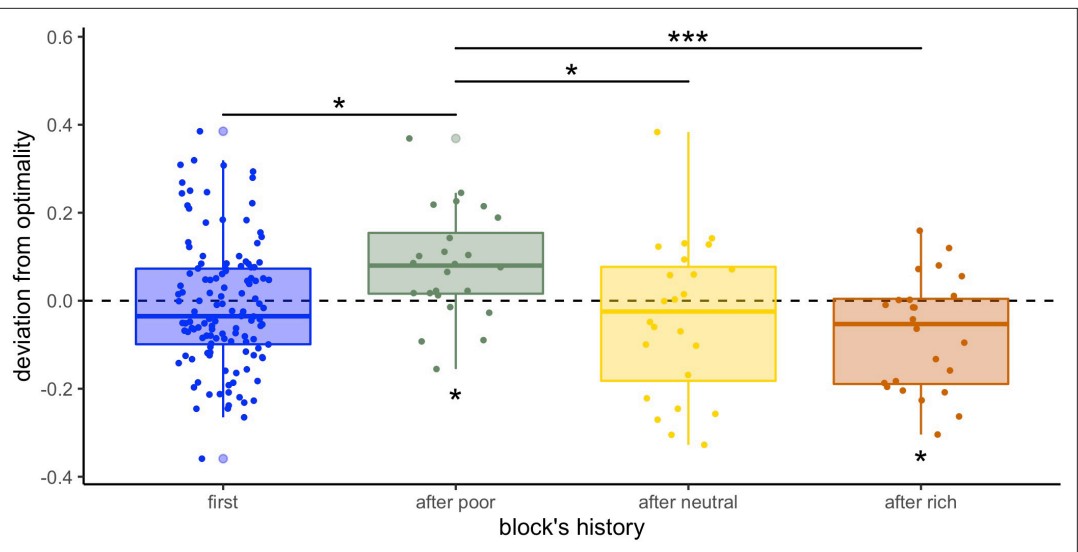

**Figure 5.** Participants' sampling strategy is affected by the previous environment presented. Distribution of the individual deviation from optimality (difference between power factors $a$ fitting the data and the optimal BD trade-off) for each block's history (presented first or independently, or after another environment). '*': p<0.05, '**': p<0.01, '***': p<0.001. Lower and upper hinges correspond to the 1st and 3rd quartiles and vertical lines represent IQR*1.5. Sample sizes: 'first': n=126, 'after poor/neutral/rich': n=24.

The online version of this article includes the following figure supplement(s) for figure 5:

**Figure supplement 1.** Participants' sampling strategy seems to be influenced by the previous environment presented.

**Figure supplement 2.** Participants tend to be closer to optimal in the second compared to first part of the block but contamination effects due to the environment change do not fully dissipate.

*Figure 4—figure supplement 1*). We observed that participants' strategy significantly shifted towards the optimal regime from the first to the second half of the block, in the poor ($V = 615$, $p_{adj} = .0085$) and the rich environment ($V = 1651$, $p_{adj} = .0015$), but not in the neutral environment ($V = 1271$, $p_{adj} = .88$). Therefore, through experience within a block, participants become closer to optimal. The magnitude of this improvement was not significantly different in the rich compared to the poor environment ($W = 2222$, $p = .84$), suggesting that the amount of reward accumulated doesn't influence performance.

Finally, we wondered whether the above deviations from optimality were more pronounced in the within-subject designs (W10 and W32), where it is possible that experience on one environment carried over the next experienced environment. We observe that in several cases sampling behaviours seem to shift towards breadth in environments directly following the presentation of a poorer environment, or towards depth if a richer environment was presented immediately before (*Figure 5—figure supplement 2*). For instance, participants' sampling strategy in the neutral environment (*Figure 5—figure supplement 1* middle panels) seems to differ depending on whether it was presented first or not. If presented after the poor environment, we observe a clear deviation towards breadth, whereas a shift towards depth is observed if presented after the rich environment. To statistically test the presence of a sequential, or contamination effect, on participants' sampling strategy, we compared the exponent estimated from fitting the number of observed and optimal alternatives sampled $M$ depending on the capacity using the power-law model ($P$) previously described (*Figure 5*). We observed significant positive deviations from optimality in the power factor after the presentation of a poor environment (deviation mean ± s.d.: 0.08±0.12, one-sample t-test, $t_{23} = 3.26$, $p_{adj} = .014$), suggesting that participants sample with more breadth when previously presented with a poor environment. In contrast, we observed negative deviations after the presentation of a rich environment (–0.08±0.13, $t_{23} = -2.85$, $p_{adj} = .036$), suggesting that participants sample more deeply when previously presented with a rich environment. Blocks presented first (in within-subject designs or independently in between-subject designs) or after a neutral environment do not significantly deviate from optimality (first: –0.01±0.14, $t_{126} = -0.81$, $p_{adj} = 1$, after

neutral: –0.04±0.18, $t_{23} = -1.13$, $p_{adj} = 1$). Overall, we observe a significant effect of block history on participants' sampling strategy (measured by the difference between the observed and the optimal exponent: $a_{observed} - a_{optimal}$) (ANOVA, $F_3 = 5.23$, $p = .002$). To confirm this, post hoc analyses revealed that participants' sampling strategies shifted towards depth in blocks presented after a poor environment compared to blocks presented after a neutral (t-test with Bonferroni correction: $t_{24} = 2.79$, $p_{adj} = .048$) and rich environments ($t_{24} = 4.32$, $p_{adj} = 5.1 \times 10^{-4}$), and compared to blocks presented first ($t = -3.27$, $p_{adj} = .014$). No significant shift was found after being presented to a rich environment compared to blocks presented first ($t = 2.21$, $p_{adj} = .20$). Although based on exploratory analyses, these observations provide some evidence that participants' sampling strategy was affected by the previous environment presented and that adaptation to a new environment requires to overcome the behavioural pattern implemented at earlier blocks.

To follow up on this history effects we further investigated if this overcoming from the previous strategy happens totally, partially or at all within the timescale of a block. We divided each block in two halves as a function of trial number (median split) and submitted the data to an ANOVA with experience (first or second block half) and environment context (poor, neutral, or rich) inside both block history types (with an environment change or not). First, we found a significant interaction between block half and environment in both cases (ANOVAs, blocks no-change: $F_2 = 5.29$, $p = .0063$, change: $F_2 = 3.95$, $p = .024$) showing that participants seem to be closer to optimal in the second compared to the first half of the block (post hoc comparisons are reported in **_Supplementary file 2_** and in **_Figure 5—figure supplement 2A_**). Additionally, when running an ANOVA over the whole data with experience, environment, and block history, we don't observe any significant three-way interaction ($F_3 = 0.76$, $p = .52$), suggesting that the magnitude of this improvement is not statistically different depending on the block's history. However, the contamination effects observed in blocks presented after another environment (environment change) do not seem to dissipate as a function of experience (time) within a block, because the block's history (presented first or after a poor, neutral, or rich environment) still has a significant effect on the power factor $a$ deviation from optimality ($a_{observed} - a_{optimal}$ in the second half of the block (median split on the number of trials, ANOVA: $F_3 = 2.95$, $p = .039$, post

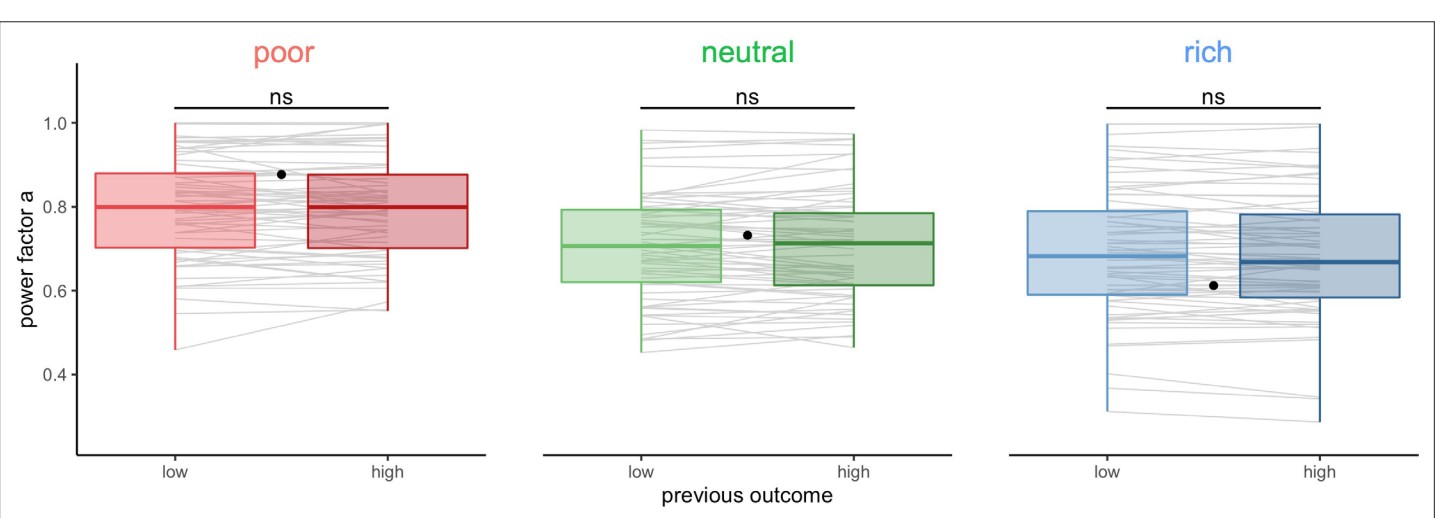

**Figure 6.** Participants' sampling strategy is not affected by the outcome obtained in the previous trial. Distribution of the power factor $a$ in the power-law model when fitting the number of alternatives sampled $M$ as a function of the capacity in each environment, depending on the magnitude of the reward obtained in the previous trial (median split on the trial reward inside capacity and environment conditions). Each line connects a subject, black dots represent the power factor $a$ when fitting the optimal breadth-depth (BD) trade-offs. Results of post hoc comparisons are displayed according to adjusted p-values ('ns': $p_{adj} > 0.1$). Lower and upper hinges correspond to the 1st and 3rd quartiles and vertical lines represent IQR*1.5. Sample sizes per environment condition: n=66.

The online version of this article includes the following figure supplement(s) for figure 6:

**Figure supplement 1.** Participants' sampling strategy is not affected by the outcome obtained in the previous trial.

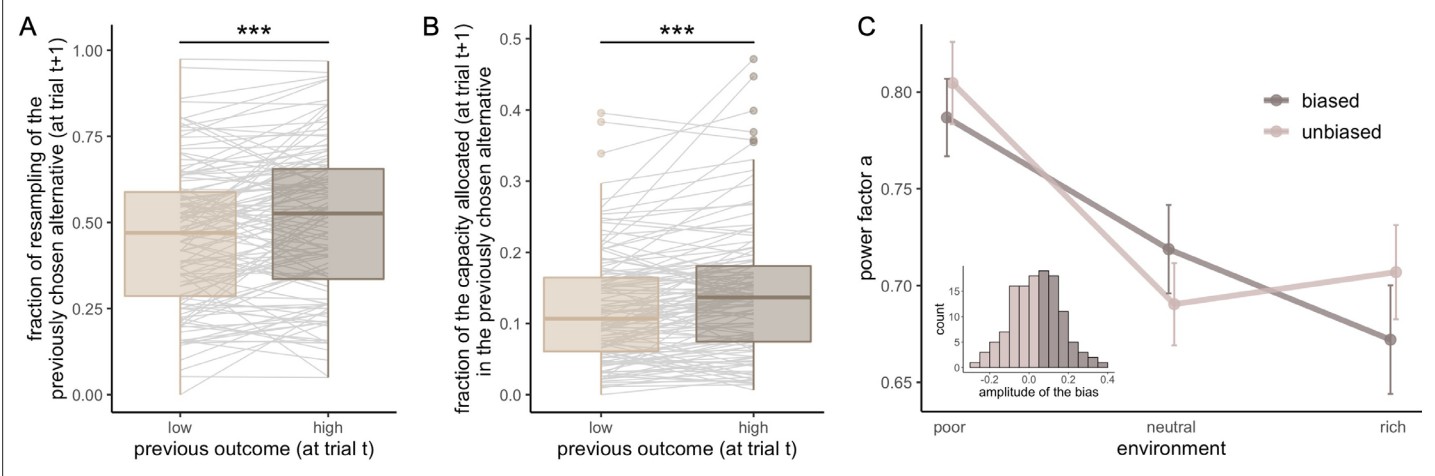

**Figure 7.** Participants resample more often and with more samples the previously chosen alternative when it was associated with high reward but it has no impact on the breadth-depth (BD) trade-off adaptation to the environment richness. Fraction of trials for which the previously selected alternative (at trial *t*) was sampled on the consecutive trial (at trial *t+1*) (resampling fraction) (**A**) and fraction of the capacity allocated in this alternative (**B**), depending on its previous associated outcome (at trial *t*, median split: low and high) overall. Grey lines connect individual data. '***': p<0.001. Lower and upper hinges correspond to the 1st and 3rd quartiles and vertical lines represent IQR*1.5. Sample size: n=126. (**C**) Averaged power factors extracted from fitting a linear model to values *M* vs. capacity in a log-log scale for participants showing a preference to resample previously rewarding alternatives (≥5%, 'biased') or not ('unbiased'). Errors bars represent s.e.m. Sample size for each experiment paradigm for the biased (B10: 23, W10: 10, B32: 15, W32: 12) and unbiased participants (B10: 22, W10: 8, B32: 30, W32: 6).

The online version of this article includes the following figure supplement(s) for figure 7:

**Figure supplement 1.** Participants' resampling bias for highly rewarded alternatives is not found at the environment level.

hoc comparisons are reported in *Supplementary files 3 and 4* and in *Figure 5—figure supplement 2B*). Therefore, the carryover effects from the previously used strategy are long-lived.

### Short-term sequential effects are absent or weak

The above long timescale effects could be the result of learning or adaptation mechanisms that act on a much shorter timescale and that builds up or wanes over time. Therefore, we studied short-term effects of the outcome and choices in one trial on the choices made by the participants in the following trial. In particular, the outcome on the previous trial might affect the BD trade-offs in the subsequent trial. We classified trials within each block depending on the reward obtained in the previous trial (median split, controlled for capacity) and again tested the individual BD trade-offs by fitting the power-law model (*Figure 6*, BD trade-offs are presented in *Figure 6—figure supplement 1*). We found that participants' sampling strategy was not significantly different following a low or high reward trial (paired Wilcoxon tests, poor: $V = 1065$, $p_{adj} = 1$, neutral: $V = 1154$, $p_{adj} = 1$, rich: $V = 1324$, $p_{adj} = .50$).

We then further investigated the impact of previously rewarded alternatives on the sampling in the next trial. We again split trials depending on the magnitude of the reward obtained in the previous trial (high vs. low median split, corrected for capacity). We observed a tendency for participants to resample (allocate at least one sample) the previously chosen alternative more often after high reward than low reward (probability mean ± s.d. low: 0.46±0.21, high: 0.50±0.22, paired t-test, $t_{125} = -3.5$, $p = 6.54 \times 10^{-4}$, *Figure 7A*). The size of this effect is small (Cohen's d: $d = .31$) and is not significant when tested inside each environment individually (paired Wilcoxon tests with Bonferroni correction, poor: $V = 1829.5$, $p = .34$, neutral: $V = 1956$, $p = .94$, rich: $V = 1911$, $p = .68$, *Figure 7—figure supplement 1A*). A similar bias was observed when considering, at trial *t+1*, the fraction of samples allocated in the previously selected alternative, at trial *t* (corrected for the capacity available) (*Figure 7B*). Participants resampled more (allocate more samples to) the previously chosen alternative when it had been associated with a high reward (median split, mean ± s.d. low: 0.12±0.08, high: 0.14±0.09, paired Wilcoxon test, $V = 2215$, $p = 1.39 \times 10^{-5}$). As before, the size of this effect was not large (Cohen's d: $d = .41$) and did not survive when testing each environment individually (paired

Wilcoxon tests with Bonferroni correction, poor: $V = 1793$, $p = .24$, neutral: $V = 1956$, $p = .94$, rich: $V = 1947$, $p = .88$, *Figure 7—figure supplement 1B*).

Despite the previous analysis showed, overall, that the magnitude of the reward obtained in the previous trial influences whether and how much the previously chosen alternative will be sampled in a subsequent trial, this bias does not affect sampling strategies (ANOVA: $F_1 = .11$, $p = .74$) nor interacts with the effect of environment richness (between: $F_1 = .74$, $p = .39$, within: $F_1 = .50$, $p = .48$). It is not suboptimal either, as rewards are randomly assigned to each location at every trial. What is more, the main effect of interest, environment richness on the BD trade-off, remains significant (between: $F_1 = 7.43$, $p = .0074$, within: $F_1 = 33.71$, $p = 1.72 \times 10^{-7}$) even when run only on participants displaying the bias (difference in the fraction of resampling between high and low outcome associated alternative ≥0.05, N=60, between: $F_1 = 4.98$, $p = .030$, within: $F_1 = 18.17$, $p = 1.08 \times 10^{-4}$ , *Figure 7C*).

We also detected preferences for sampling alternatives associated with slightly less costly motor actions (see *Appendix 3—figure 1*) but, as those previously associated with high reward, we have no evidence showing that these biases affect optimality.

## Model comparison

To closer characterize the behavioural patterns, we generated quantitative predictions from a variety of models, including the optimal model and several heuristics, and tested which are better able to predict the empirical observations. We chose to represent extreme behaviours, such as full-breadth ($M$ is always equal to capacity $C$) or depth ($M$ always equal to 2) as boundary models, as well as in-between behaviours which would represent the use of a trade-off between breadth and depth. Based on observations of the data and predictions (*Moreno-Bote et al., 2020*), we chose to model $M$ as function of $C$ using linear and power-law models. In the latter case, we used two versions of the power-law model: one where $M$ increases with the square root of capacity (exponent of 1/2) (*Moreno-Bote et al., 2020*), and another where the exponent was free to vary (see Materials and methods for details of the models). To model realistic behavioural patterns, we injected noise into the model predictions that corrupts the chosen number of suppliers $M$. As noise models, we used binomial and discretized Gaussian distributed noise added to the model prediction of $M$. In the second case, the noise follows a normal distribution with a standard deviation that increased linearly with capacity. This choice was made based on the empirical observation that the variance of $M$ increases linearly with capacity (Kendall's rank correlation test, $z = 11.26$, $tau = .87$, $p < 2.2 \times 10^{-16}$ , see *Figure 8—figure supplement 1*). Data from the four experimental designs were analysed and are presented together. Fourfold cross-validated log-likelihoods (CVLL) are used to compare the models (Materials and methods; qualitatively identical results hold when using Akaike information criteria (AIC) instead, see *Figure 8—figure supplements 3 and 4*). Using binomial noise, the free power-law model outperforms all five other models (*Figure 8A* and *Table 3*). The optimal model, even though less efficient than the power model, is better at predicting participants' sampling strategy than the pure breadth, depth, and square root models. The outcomes of model comparison are mostly equivalent when using the Gaussian noise model (for the results of this model, see *Figure 8—figure supplement 2* and *Supplementary file 5*), and therefore we restrict our discussion to the binomial noise model from now on. We just note that the only divergence between the noise models appeared in the pair-wise comparisons, with the hierarchy of models' performance being more clearly statistically established in the binomial noise model compared to the Gaussian one.

We observed a significant interaction between the environment and the models on the CVLL (Scheirer-Ray-Hare test; $H_5 = 24.70$, $p = 1.59 \times 10^{-4}$). The depth model is better at predicting the data in the rich compared to the poor environment (Wilcoxon test with Bonferroni correction, $W_{66} = 348$, $p_{adj} = 3.99 \times 10^{-6}$). The square root model is also shown to perform worst in the poor environment compared to the rich ($W_{66} = 379$, $p_{adj} = 1.06 \times 10^{-5}$) and the neutral ($W_{66} = 325$, $p_{adj} = 1.88 \times 10^{-6}$) environments. Additionally, the free power-law model once again outperforms any other models in all environments (*Figure 8B*).

As overall the free power-law model is best at predicting the empirical data, we chose to confirm the effect of the environment on participants' sampling strategy (see *Figure 3*) comparing the exponents parameters estimated from the power-law model between environments (*Figure 8C*). This new analysis confirmed the results reported earlier (see *Figure 3*), with a significant effect of the environment on the exponent $w$ in all designs B10 (Kruskal-Wallis test, $K - W \chi_2^2 = 14.43$, $p = 7.34 \times 10^{-4}$), W10

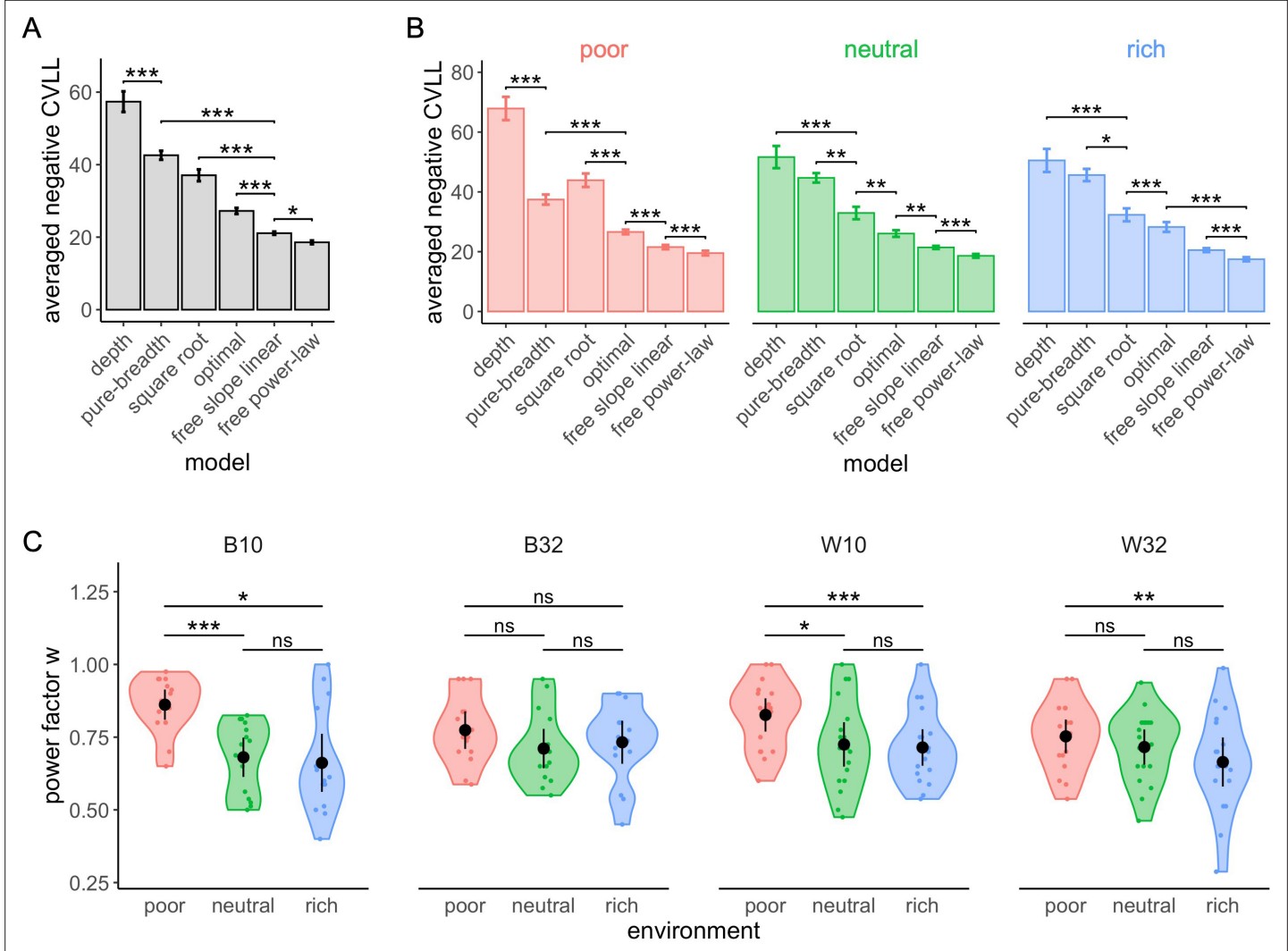

**Figure 8.** The free power-law model is better at predicted participants' sampling strategy than both the optimal and other heuristic models (using binomial distributed noise). Averaged negative cross-validated log-likelihood (CVLL) across participants for each model overall (**A**) and in each environment (**B**). Results of pair-wise comparisons are displayed according to adjusted p-values ('*': p<0.05, '**': p<0.01, '***': p<0.001) and error bars correspond to s.e.m. Sample sizes (**A**): n=126, (**B**): n=66 for each environment condition. (**C**) Distribution of the power factor $w$ in the free power model. Each colour dot represents a subject, black dots represent distribution means, and bars 95% confidence intervals. Results of post hoc comparisons are displayed according to adjusted p-values ('ns': p>0.05, '**': p<0.01, '***': p<0.001). Samples sizes per environment condition: designs B10 and B32: n=15, designs W10 and W32: n=18.

The online version of this article includes the following figure supplement(s) for figure 8:

**Figure supplement 1.** The standard deviation of the number of alternatives sampled $M$ increases linearly with the capacity.

**Figure supplement 2.** The free power-law model is better at predicting participants' sampling strategy than both the optimal and other heuristics models using Gaussian distributed noise.

**Figure supplement 3.** The free power-law model is better at predicting participants' sampling strategy than both the optimal and other heuristics models using binomial distributed noise.

**Figure supplement 4.** The free power-law model is better at predicting participants' sampling strategy than both the optimal and other heuristics models using Gaussian distributed noise.

(one-way ANOVA, $F_1 = 26.47$, $p = 8.11 \times 10^{-5}$), and W32 ($F_1 = 12.81$, $p = .002$), except for the design B32 ($F_1 = 0.84$, $p = .37$). In all three designs W10, B10, and W32, we observed significant differences in the power exponent between the poor and the rich environment (t-tests with Bonferroni corrections, W10: $t_{17} = 5.15$, $p_{adj} = 2.43 \times 10^{-4}$, B10: $W_{15} = 182$, $p_{adj} = .012$, W32: $t_{17} = 3.58$, $p_{adj} = .007$).

**Table 3.** Summary of the pair-wise comparisons (Wilcoxon matched pairs signed-ranks test) of the fourfolds averaged cross-validated log-likelihoods (CVLL) between all six models using binomial distributed noise.

p-Values are adjusted with Bonferroni corrections and significative differences (p<0.05) are highlighted in bold. Models are ordered from worst (depth) to best (free power law).

| | Pure breadth | Square root | Optimal | Linear | Power |
|---|---|---|---|---|---|
| Depth | $V_{126} = 2796$, $p_{adj} = .051$ | $V_{126} = 2$, $\boldsymbol{p_{adj}} = 3.22 \times 10^{-21}$ | $V_{126} = 176$, $\boldsymbol{p_{adj}} = 1.91 \times 10^{-19}$ | $V_{126} = 65$, $\boldsymbol{p_{adj}} = 1.44 \times 10^{-20}$ | $V_{126} = 2$, $\boldsymbol{p_{adj}} = 6.90 \times 10^{-21}$ |
| Pure breadth | | $V_{126} = 2926$, $p_{adj} = .134$ | $V_{126} = 1154$, $\boldsymbol{p_{adj}} = 6.35 \times 10^{-11}$ | $V_{126} = 4$, $\boldsymbol{p_{adj}} = 4.95 \times 10^{-21}$ | $V_{126} = 2$, $\boldsymbol{p_{adj}} = 3.22 \times 10^{-21}$ |
| Square root | | | $V_{126} = 1243$, $\boldsymbol{p_{adj}} = 2.86 \times 10^{-10}$ | $V_{126} = 746$, $\boldsymbol{p_{adj}} = 3.48 \times 10^{-14}$ | $V_{126} = 14$, $\boldsymbol{p_{adj}} = 1.98 \times 10^{-20}$ |
| Optimal | | | | $V_{126} = 1510$, $\boldsymbol{p_{adj}} = 2.01 \times 10^{-8}$ | $V_{126} = 143$, $\boldsymbol{p_{adj}} = 8.92 \times 10^{-20}$ |
| Linear | | | | | $V_{126} = 259$, $\boldsymbol{p_{adj}} = 4.49 \times 10^{-18}$ |

We also observed significant differences between the poor and the neutral environment in the designs W10 ($t_{17} = 3.16$, $p_{adj} = .017$) and B10 ($t_{15} = 200$, $p_{adj} = 8.37 \times 10^{-4}$) but not in the design W32 ($t_{17} = 2.14$, $p_{adj} = .142$). Moreover, we did not find any significant difference in the power exponent $w$ values between the neutral and rich environments in any of the designs (W10: $t_{17} = 0.42$, $p_{adj} = 1$, B10: $t_{15} = 127$, $p_{adj} = 1$, W32: $t_{17} = 1.62$, $p_{adj} = .37$).

## Participants tend to sample homogeneously amongst alternatives

Above, we have presented analyses regarding how participants allocate the samples within the alternatives selected during sampling and compared it to optimal allocation of samples across capacities and environments. Deviations from the optimal strategy are clear, as shown both in the empirical data (which often deviates from the optimal allocation) and in the outcome of model comparison to fit these empirical data. This deviation is interesting because the difference from optimality is not exclusively accounted for by unbiased noise introduced in the model comparison above, suggesting the existence of a source of systematic bias in human choice behaviour.

In order to better understand these deviations from optimality, we looked in more detail at the most frequent observed allocation of samples. We observed that participants frequently allocated the samples homogeneously across the selected alternatives, a sampling strategy which in most cases differs largely from optimal allocation (*Figure 9A*). To characterize this bias towards homogenous sampling, we computed the standard deviation of the ordered counts for each allocation of samples averaged for each participant and environment (Materials and methods) and compared it to the standard deviation of the optimal allocation (*Figure 9B*). We observed that the standard deviation of the sample allocations measured in the poor (Wilcoxon test with Bonferroni correction, $W_{66} = 529$, $p_{adj} = 7.01 \times 10^{-4}$), neutral ($W_{66} = 295$, $p_{adj} = 6.87 \times 10^{-7}$), and rich ($W_{66} = 48$, $p_{adj} = 4.38 \times 10^{-11}$) environments were all significantly lower than the standard deviation assuming optimal sample allocation. Moreover, we observed that the magnitude of these differences differed between environments (Kruskal-Wallis test, $K - W \chi^2_2 = 111.92$, $p < 2.2 \times 10^{-16}$). In fact, they increased with the environment richness, with larger differences observed in the rich compared to neutral (Wilcoxon test with Bonferroni correction, $W_{66} = 4089$, $p_{adj} = 1.04 \times 10^{-17}$) and poor ($W_{66} = 4183$, $p_{adj} = 2.20 \times 10^{-19}$) environments and larger differences in the neutral compared to the poor environment ($W_{66} = 2922$, $p_{adj} = .002$). That is, there was an overall bias towards homogenous sampling, and this bias was stronger as the environment was richer.

Finally, we investigated whether the observed deviations from the optimal sample allocation impacted the average reward obtained by the participants in our task. In each trial, the reward or outcome is defined as the number of good-quality apricots among the 100 apricots bought. When computing the differences between the observed and optimal outcomes (the averaged outcome obtained when following the ideal observer), we observed significantly negative deviations from optimality in all three

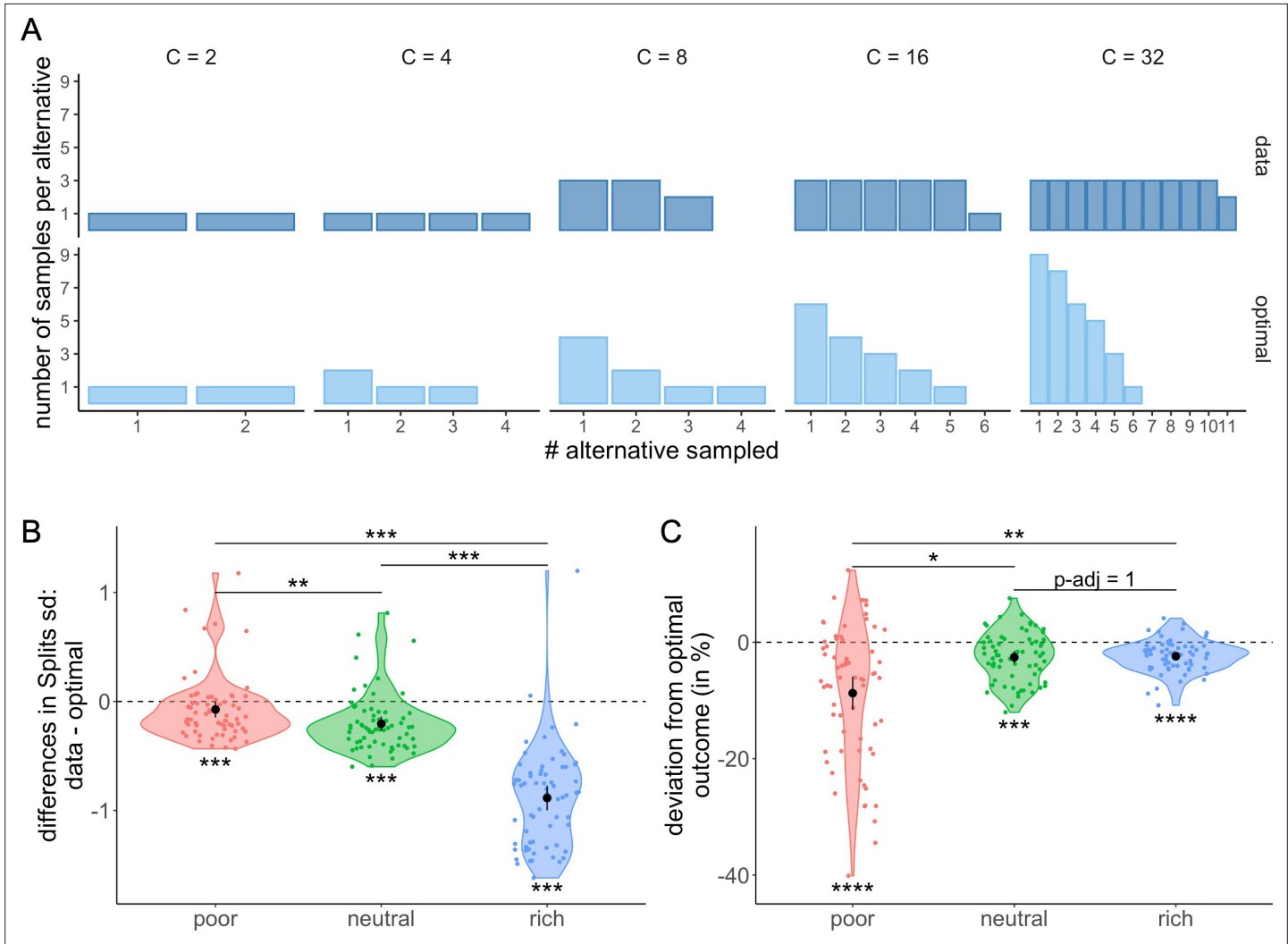

**Figure 9.** Participants tend to allocate homogeneously the samples across the selected alternatives, which largely differs from the optimal allocation, and impacts the outcome. (**A**) Number of samples allocated to each sampled alternative depending on the sampling capacity $C$. Upper panels: most frequent allocation of samples observed across participants in the rich environment (design B32) as a function of capacity. Lower panels: allocation of samples maximizing the reward (optimal). (**B**) Distributions of the differences between observed and optimal standard deviations of the distribution of samples among the selected alternatives in each environment (e.g. if $C = 4$ and 2 samples are allocated in a first alternative while the last 2 samples are each allocated in a second and third alternative, the standard deviation of this sample allocation would correspond to $\mathrm{sd}\left(\{2, 1, 1\}\right) \approx 0.577$). Note that more homogeneous distributions tend to lead to lower standard deviations. (**C**) Distributions of the differences between observed and optimal outcomes in each environment. In the last two panels, dots represent participants and include all trials for which the optimal number of alternatives sampled was inferior to capacity ($M_{opt} < C$ – see Materials and methods for more details). Below each distribution are presented results of one-sample Wilcoxon tests ('**': $p_{adj}$ <0.01, '***': $p_{adj}$ <0.001) and above are presented results of Wilcoxon tests between each environment ('ns': $p_{adj}$ >0.05, '*': $p_{adj}$ <0.05,'**': $p_{adj}$ <0.01,'***': $p_{adj}$ <0.001). All p-values have been adjusted with Bonferroni corrections. Sample sizes for each environment condition: n=66.

environments (mean ± s.d. in poor: –8.74 ± 11.60%, neutral: –2.56 ± 4.23%, and rich: –2.39 ± 2.74%; two-way Wilcoxon tests with Bonferroni correction, poor: $W_{66} = 311$, $p_{adj} = 1.18 \times 10^{-6}$, neutral: $W_{66} = 457$, $p_{adj} = 1.04 \times 10^{-4}$, rich: $W_{66} = 228$, $p_{adj} = 6.33 \times 10^{-8}$) (**Figure 9C**), showing that the bias observed in the sample allocation did have a deleterious impact on the outcome. The magnitude of these differences between optimal and actual outcome was modulated by the environment (Kruskal-Wallis test, $K - W \chi_2^2 = 12.29$, $p = .002$), and while the harmful effect of a suboptimal strategy was moderate in the poor environment, it was very small in the remaining two. Surprisingly, the direction of the effect of environment on outcome was opposite to the one observed on standard deviations, as more deviation from optimality in the rich environment did not affect performance as much as in

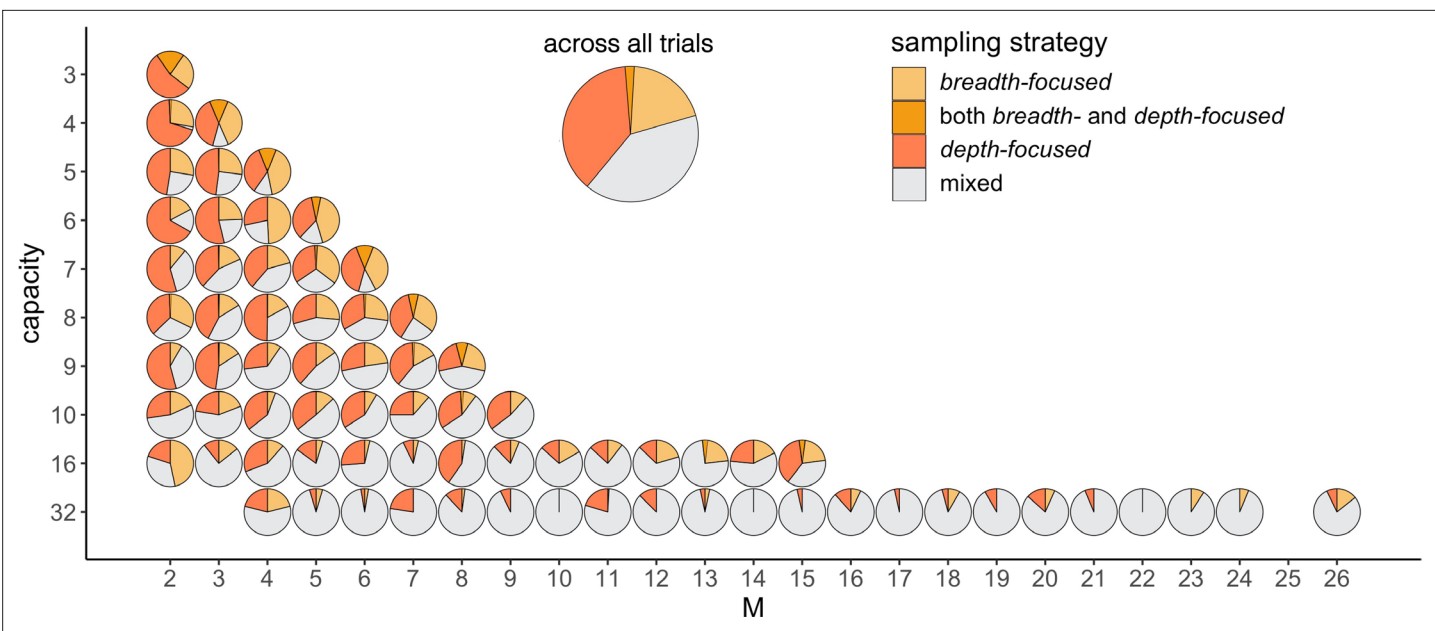

**Figure 10.** Participants prefer the depth- over the breadth-focused strategy. The fraction of trials for which alternatives are sampled according to the depth- (dark-orange) vs. breadth-focused (light-orange) strategy is shown for each number of sampled alternatives ($M$) and capacity ($C$). Example of a sampling sequence of alternatives {a,b,c,d} with $C = 7$ and for $M = 4$ in breadth-focused strategy: {a,a,b,c,c,d,d} and breadth-focused strategy: {a,b,c,d,d,a,c}. This analysis includes all trials for which $M > 1$ (no pure-depth) and $M < C$ (no pure-breadth). Only combinations of $M, C$ with at least 10 trials (over all participants) were displayed.

the poor environment. This inverted correlation is due to the nature of the environment itself. Indeed, a small deviation from the optimal strategy in the poor environment has a much larger impact on the outcome than in the rich environment, where a great majority of alternatives have a high probability of success. Indeed, we observed that the difference between the optimal and the observed outcomes seemed to decrease as the environment richness increased with differences in the poor environment being significantly higher than the ones in the neutral (Wilcoxon tests with Bonferroni correction, $W_{66} = 1549$, $p_{adj} = .013$) and rich environment ($W_{66} = 1475$, $p_{adj} = .004$). No significant difference was observed in between the neutral and rich environments ($W_{66} = 2185$, $p_{adj} = 1$).

Participants' tendency to sample homogenously may also be one of the sources of deviation from optimality observed in the number of alternatives sampled $M$. Indeed, we observe that among the trials where participants did not use a pure breadth nor a pure depth strategy ($M < C$ and $M > 1$), they first allocate all the samples in an alternative before sampling another one (*depth-focused*) on 39.9% of the trials. In contrast, they chose to first sample once all the alternatives $M$ before adding additional samples over them (*breadth-focused*) on 21.8% of the trials. While the first strategy prioritizes the number of samples per alternative, the second one focuses on the number of alternatives sampled. We investigated further how these strategies prevail depending on $C$ and $M$ and observed that while the breadth-focused strategy is mostly present for trials with a shallow sample allocation ($M$ close to $C$), the depth-focused strategy is present across all types of BD trade-offs ($M/C$ ratios) (*Figure 10*).

## Optimal or close-to-optimal sampled alternatives are mostly chosen

In the normative model the optimal sampled alternative is the alternative $i$ that maximizes the normative value $V_{norm}^i = \frac{(\sum O_{s,i} + \alpha)}{N_i + \alpha + \beta}$, where $\alpha$ and $\beta$ are the parameters describing the prior distribution of rewards in the current environment. We observed that participants select the optimal sampled alternative according to the above rule on $96.0 \pm 4.34\%$ (mean ± s.d.) of the trials. If we consider a different decision rule that selects the option with the highest proportional outcome $V_{prop}^i = \frac{\sum O_{s,i}}{N_i}$ (independently of the environment and thus independently of the parameters $\alpha$ and $\beta$), we observe that participants select the best sampled alternative on $96.6 \pm 4.11\%$ (mean ± s.d.) of the trials. This is significantly larger than when considering the normative optimal alternative (*Appendix 4—figure 1*), paired Wilcoxon test, ($V = 454$, $p = 6.83 \times 10^{-4}$), although very small in magnitude. Further, this effect

is not homogenous amongst all environments (permutation ANOVA, $F = 4.03$, $p = .020$) and is only significant in the poor (paired Wilcoxon test with Bonferroni correction, $V = 87.5$, $p_{adj} = 5.82 \times 10^{-4}$) and rich environments ($V = 14.5$, $p_{adj} = 4.68 \times 10^{-4}$). No significant difference is found in the neutral environment ($V = 175$, $p_{adj} = 1$). These results may also reflect difficulties for participants to estimate the relative richness of the environmental context.

In few cases (3.40 ± 4.11% of trials) when participants did not select the highest proportional alternative (the one maximizing the proportional outcome $V_{prop}$), we analysed what was the number of samples allocated in the selected alternative. We observe that, in such cases, the selected alternative had significantly more samples than the alternative maximizing the proportional outcome (mean ± s.d. of the difference in samples number: 1.78±1.51, Wilcoxon test: $V = 1303$, $p = 2.02 \times 10^{-9}$), suggesting that participants favour less uncertain alternatives.

## Discussion

Finding the best way to balance opposite strategies is a ubiquitous problem in decision making. Dilemmas such as speed or accuracy (**Fitts, 1966**; **Wickelgren, 2016**), exploration or exploitation (**March, 1991**), breadth or depth, are zero-sum situations. Increasing one necessary comes to the detriment of the other, and maximizing overall expected utility requires to strike an optimal trade-off between them. For instance, in the classic EE dilemma exploiting a known reward option precludes exploring potentially better ones. The BD dilemma, studied here, is related to EE but has the crucial difference that a limited resource can be divided into several options simultaneously. Therefore, both exploration and exploitation could a priori be performed in parallel and interact with each other. The trade-off that arises in BD is twofold. First, given limited capacity, how much of it should be used for exploration and how much for exploitation. And second – and as we chose to focus on in this paper – during exploration the agent can further divide finite resources to learn much about few alternatives or little about many of them. These trade-offs do not naturally arise in the EE, as it is the divisibility of the agent's resource, and thus a new degree of freedom, what results in a qualitatively new problem. By focusing on the latter, here we provide answers on how human participants manage the allocation of finite, but potentially large, resources.

Previous research has formalized the BD dilemma using rational decision theory under capacity constraints (**Mastrogiuseppe and Moreno-Bote, 2022**; **Moreno-Bote et al., 2020**; **Ramírez-Ruiz and Moreno-Bote, 2022**). This research has described the optimal allocation of capacity over multiple alternatives as a function of both agent's capacity and environmental richness. At low capacities, it is optimal to draw one sample per alternative (pure-breadth strategy), while for larger capacities, some alternatives are ignored (balancing breadth and depth), and the number of alternatives sampled roughly increases with the square root of capacity, independently of the richness of the environment. Additionally, the passage from pure breadth to a BD trade-off occurs with a sharp transition at a specific capacity which directly depends on the overall success probability of the alternatives. Indeed, the transition is shifted towards larger capacity values as the environment gets poorer. Finally, it is optimal to favour uneven allocations of samples among the considered alternatives, especially as capacity and environment richness increases. Rational decision theory provides us with a normative hypothesis about how humans ought to behave.

Experimentally, BD trade-offs have been studied outside cognitive neuroscience (**Halpert, 1958**; **Schwartz et al., 2009**; **Turner et al., 1955**), mostly by using choice menus of different nature. However, this research does not address whether allocation policies used by humans are rational nor they parametrically manipulate an agent's capacity or the difficulty of the environment. To fill this gap, here we have introduced a novel experimental paradigm that allows us to compare human empirical data with normative predictions under a BD dilemma. The BD apricot task is intuitive and easily grasped by the participants. Consistent with theoretical predictions, we observe that, at low capacity, a pure-breadth strategy is favoured (sampling as many alternatives as possible), while as capacity increases, participants progressively sample relatively fewer of them in more depth. Additionally, our results reveal that participants' sampling strategy adapts to the environment with richer environments promoting depth over breadth.

We observe that human behaviour is close to, but systematically deviated, from optimality. Regarding the number of alternatives sampled as a function of capacity, we observe that participants'

sampling strategy does not follow the predicted optimal model but is better captured by a power law. This strategy has the advantage to be continuous (no break point), which might contribute to reduce computational (hence, cognitive) cost, within a controlled loss of efficiency. Most likely is however an explanation based on the presence of sample allocation noise, which largely smooths the predicted sharp transitions between low and high capacity, noise for which we found some evidence (see *Figure 4—figure supplement 1*). We also demonstrated that, at low capacity, participants often sample deeper than optimal, whilst at high capacities the reverse pattern is observed, with participants sampling more shallowly than optimal. Additionally, the deviations towards depth at low capacities are especially observed in poorer environments, while deviations towards breadth at high capacities are more prevalent in richer environments. Given that environment richness is not known precisely by participants, such results may be explained by errors in the estimation of the environment richness (*Drugowitsch et al., 2016*; *Schustek et al., 2019*; *Wyart and Koechlin, 2016*). However, they could also have more complicated origin. *Vul et al., 2014*, showed that the optimal number of samples allocated per sampled alternative directly depends on the action/sample cost ratio in a way that a low action/sample cost ratio promotes depth over breadth. In our paradigm, the ratio between the gain in outcome (outcome received after sampling compared to chance) and the sampling cost (linearly increasing with the number of samples) is not constant over all capacities and environments. As a result, this change in ratio could also play a role in explaining why the amplitude of the deviations observed vary depending on the environment richness.

Regarding how samples are allocated across alternatives, we observe a participants' tendency to sample alternatives homogenously. This distribution of samples deviates significantly from optimal behaviour but caused only a small reduction of outcome obtained compared to optimal (mean ± s.d. from optimality: –3.5 ± 6.24%). At a cognitive level, such bias towards an even distribution of samples may have various origins. It could emerge to simplify computations during sampling due to an even division of samples, which might be easier to remember and implement as a motor output. It can also emerge to simplify comparisons during choice in at least two ways. First, it is easier to compare fractions with a common denominator, thus reducing cognitive load. Second, a homogeneous allocation of samples ensures that alternatives are equally risky, as they carry the same amount of information. Indeed, humans have shown a preference to sample (note, not choose) the most uncertain option (*Alméras et al., 2021*; *Lewis, 1995*; *Schulz et al., 2019*; *Wilson et al., 2021*), a heuristic strategy that has been called 'uncertainty directed exploration'. In our study, with delayed feedback, sampling homogenously the alternatives renders equal uncertainty about all sampled alternatives. Homogenous sampling could also have other sources, such as the human preference for symmetry (*Attneave, 1955*). We tried to infer the thought processes that participants use while sampling through the identification of patterns in the sampling sequence and thus have a better understanding of the bias towards symmetric allocation. Our results suggest that participants focus more on the number of samples per alternative over the number of sampled alternatives (see *Figure 10*). Consequently, participants' bias to sample homogenously may drive, at least in part, deviations from optimality observed in the number of sampled alternatives.

Such systematic deviations from optimality and biases can also emerge from individual traits. In our dataset, we did observe individual differences in the way participants adapt their strategy to both the resources available (capacity) and the environment richness (see *Figure 2A*) but we could not relate these differences to neither an effect of gender nor age (see section 'Individual differences' in the Materials and methods for a preliminary analysis). However, although the nature of our sample probably doesn't allow to properly study a potential effect of age (75% of our participants are under 30), we believe the BD apricot task give a great framework to specifically study how target populations manage limited search capacity. We also observed individual differences in the way participants' sampling strategies deviate from optimality (see Results section 'Deviations from optimality'). We hypothesize that deviations towards either breadth or depth may partially originate from a participants' will to reduce uncertainty about either finding a 'good' alternative during the sampling phase (one with positive outcome(s)) or correctly estimating the reward during the purchase phase. On the one hand, deep sampling enables to better estimate the success probability of the sampled alternatives in order to find the best one but at the risk of not having any good one. This strategy is more reward centred and may be followed by individuals with low risk aversion. On the other hand, sampling broadly reduces the risk not to find any good alternative. Such a strategy might be more

cautious and may be followed by individuals with higher risk aversion profiles. This hypothesis will need to be tested in future studies, by estimating separately participants' relation to risk, using self-reported risk (*Dohmen et al., 2005*) and behavioural measures of risk taking (*Eckel and Grossman, 2008*; *Holt and Laury, 1958*; *Lejuez et al., 2002*).

Finally, even though our paradigm was introduced using a real-life narrative to facilitate understanding and avoid using a mathematical or probabilistic-related vocabulary, we cannot exclude that mathematical knowledge has influenced the way participants comprehend and behave in the task. Gathering information about participants' scientific background could help us to understand better some of the deviations from optimality observed. For example, although we believe participants have a sufficient understanding of the sampled alternatives' statistics (participants select the optimal sampled alternative on more than 96% of the trials), we cannot exclude that having a better understanding of probabilities could be associated with a reduced homogenous allocation bias which may indirectly modulate the BD trade-off.

In general, even though participants engage in behaviours which deviate significantly from optimality, the heuristics identified – continuous power-law sampling strategy and homogenous sampling – are associated with a lower cognitive effort compared to the optimal behaviour. Additionally, if following these biases impact significantly the outcome, the loss remains relatively small in our task. It is not clear whether these biases reflect participants' computational limitations – bounded rationality (*Simon, 1955*) – or are the result of a compromise between the error in the sampling strategy and the cost associated with brain computations – bounded optimality (*Russell and Subramanian, 2009*) or resource rationality (*Griffiths et al., 2015*; *Lieder et al., 2012*). Importantly, none of the reported deviations can easily be explained by the presence of motor biases, as samples are allocated one by one and participants always must go to the screen centre to drag the next sample to a supplier, so that the distance to any supplier is identical and any motion direction asymmetry is the same sample to sample.

Further, despite the large number of alternatives presented in the apricot task (either 10 or 32), we did not find evidence that participants considered much fewer options than the optimal number, and therefore our study cannot be explained by – and offers a counter-example of – choice overload (*Iyengar and Lepper, 2000*; *Kuksov and Villas-Boas, 2010*; *Sethi-Iyengar et al., 2003*). Firstly, we observe that participants consider many alternatives, often more than 5 (43.3% of the trials with $C > 5$) and up to 32 in some rare cases. Secondly, although we do observe that the number of sampled alternatives is sometimes inferior to the optimal number, this is predominant at rather low capacities ($M < 8$, see *Appendix 2—figure 1*) when few alternatives are actually considered. On the contrary, at larger capacities, participants tend to consider more alternatives than optimal. In our experimental setup, no information about any alternative was directly accessible to the participant before the outcome of the allocation was revealed, which may prevent perturbations and biases in the decision process by facilitating the a priori consideration of all alternatives on an equal footing.

Our novel experimental framework has demonstrated to be appropriate for studying human decisions in the context of the DB dilemma, but it also raises some issues concerning the difficulty to test particular conditions. Concerning the effect of environment richness, we suspect contamination to happen in participants' sampling strategy when presented with different environments consecutively. The use of between-subjects designs should be favoured to thwart this effect. We also observed that the use of very large capacities ($C = 32$) is very costly for the participants and may have a negative impact on their motivation to engage in the task and the pursue of an optimal strategy. Taking into consideration these observations, our framework has the advantage to be easily adapted to investigate future questions. We are particularly curious to test whether the results observed here stand when using a, more realistic, continuous rather than a discrete capacity (*Ramírez-Ruiz and Moreno-Bote, 2022*) or investigate how participants manage to spend a limited capacity not over a single choice but over multiple ones. In addition, this design can be transposed to solve the BD dilemma in more complex choices, such as inside large decision trees (*Mastrogiuseppe and Moreno-Bote, 2022*).

To conclude, we have developed the first thorough experimental study of the BD dilemma, which, benefiting from a large adaptability and capability to translate real-life situations, offers a promising framework to study many-alternative human decision making under finite resources. Already, our results reveal the use of close-to-optimal choice behaviours which are both sensitive to the capacity

and the environment and identify some of the heuristics used to simplify computations at the cost of optimality.

## Materials and methods
### Experimental design

We developed a protocol – the BD apricot task – to study how humans sample the environment using limited resources, to make economic choices. The task was programmed using Flutter development software (flutter.dev) as an application for smartphones or tablets. This allowed testing participants outside lab premises. Participants were initially introduced with a realistic narrative that provided a concrete context to aid understanding the task goals and constraints (for a demonstration video of the task, see *Video 1*). According to this narrative, in each trial the participant had to buy an order of apricots in bulk from one specific supplier, out of many available. The goal was to maximize the amount of good-quality apricots accumulated throughout the experiment. Because suppliers vary in the proportion of good-quality apricots they serve, participants were given the opportunity to learn about the suppliers' overall quality by sampling: prior to the purchase, participants were asked to sample apricots from various suppliers of their choice using a fixed number of 'free' samples given to them. Based on this, they were to choose the supplier for the final purchase in the trial.

Specifically, each trial in the task was divided into a sampling phase and a final purchase phase. In the sampling phase (*Figure 1a*), the participant was given a number of coins (yellow dots) that varied from trial to trial. The number of coins determines the search capacity of the participant on each trial. We ran different designs, which varied in the range of available capacity throughout trials, with each capacity selected randomly and equiprobably, from the pre-established range. In low-capacity designs, the range of possible capacities ranged between 2 and 10, in steps of one, while in high-capacity designs capacity took one of the values in the set {2,4,8,16,32}. In each trial, the coins could be freely allocated one by one to any of the suppliers by clicking the active coin in the middle of the display and then by clicking the desired supplier to sample from. Participants could arbitrarily allocate the coins in a given trial (i.e. all coins to just one supplier, or each coin to a different supplier, or anything in between). The number of possible suppliers was always fixed to the maximum capacity of the design being tested. Once all the coins had been allocated (*Figure 1b*), the samples were revealed (*Figure 1c*) as either of good- (orange) or bad-quality (purple) apricots. The sampling outcomes $X_i$ at each supplier $i$ (given the range 1–10, or 1–32) followed a binomial distribution $X_i \sim B\left(n_i, p_i\right)$, where $n_i$ is the number of samples allocated in supplier $i$ and $p_i$ is the fraction of good-quality apricots in that supplier. While $n_i$ is chosen by the participants, $p_i$ is unknown to them. Based on the information collected, participants could estimate $p_i$, and based on the estimation could choose amongst the sampled suppliers (and only the sampled ones) to perform a final bulk purchase of 100 apricots. The number of good-quality apricots contained in the purchase was revealed (*Figure 1d*), and the next trial (purchase cycle) started. The cumulative sum of good-quality apricots collected, as well as a bar informing participants about their progress, were displayed at the bottom of the screen throughout the experiment.

Independently in each trial and for each supplier $i$, the fraction $p_i$ of good apricots was randomly drawn from a beta distribution (with parameters $\alpha$, $\beta$). We considered three different environments, varying in the relative abundance of good apricots ($p_i$), denoted poor ($\alpha=1/3$, $\beta=1$), neutral ($\alpha=3$, $\beta=3$), and rich ($\alpha=1$, $\beta=1/3$). Participants were either presented to one environment throughout the experiment (between-subject designs) or to the three environments in different blocks of the same experiment with block order counterbalanced between participants (within-subject designs – see below for more details). In all cases, participants were verbally instructed about the relative richness of the environment they are in (poor/neutral/rich: 'a majority of suppliers have a low/average/high proportion of good-quality apricots') and they are aware that even though alternatives are different in each trial, they are extracted from the same environment.

Four experimental designs were run varying in the number of environments presented and the sampling capacity (see *Table 1* for summary). In the two between-subject (B) designs, participants were presented only with one environment (rich, neutral, or poor), and with one of two capacities, narrow capacity (up to 10 – design B10) or wide capacity (up to 32 – design B32). In the within-subject (W) designs, participants were presented with all three environments (in blocks), either

with narrow capacity (up to 10 – design W10) or with wide capacity (up to 32 – design W32). In all designs, participants were presented with eight repetitions of each sampling capacity for each environment (one or three depending on the design). All participants were first presented with a practice block composed of 10 trials covering the whole range of sampling capacities in their design (these trials were excluded from the analyses). In the middle of the practice block, participants had to take a short quiz to assess their understanding of the task. They were then provided with feedback and in case of an insufficient score (less than six correct answers out of eight questions), they had to redo the quiz for their participation to be accepted. The whole experiment was self-paced, and opportunities were given to participants to rest in halfway through and after each block.

## Participants

Participants were recruited through the online platform Prolific ( 2021 Prolific, https://www.prolific.co/), with the criteria of being fluent in English, and with age between 18 and 52 years of age. They were asked to perform the task using a touch screen device (smartphone or tablet for designs with 10 alternatives, only tablet for designs with 32 alternatives). Participants received a fixed monetary compensation of 8€ per hour and, to increase their motivation, an additional reward was attributed depending on their final score at the task. In between-subject designs, participants who obtained the first and second top scores in each environment received respectively 20€ and 10€. In within-subject designs, the players with the three top scores overall were rewarded with 20€, 10€, and 5€, respectively.

Participants were recruited until completing a valid final sample size of 45 participants in each of the between-subject designs (B10 and B32 – 15 participants for each environment), and 18 participants in each of the within-subject designs (W10 and W32). The final sample size included in the study was 126 (84 males, mean age ± s.d.: 25.7±6.9).

Sample size was decided before starting data collection. In between-subject designs, sample size was calculated (using GPower, *Faul et al., 2009*) to achieve a 95% power in discriminating participants strategy between environments and was based on the expected effect size (estimated at Cohen's $d$=0.467) of the environment on the sampling strategy ($M \sim C$). This effect size was calculated based on the expected effect of the environment following the ideal observer (see below for more details) and the variance observed within each group (s.d.=0.3) in a previous pilot study. In within-subjects design, the expected effect size was harder to estimate due to a likely contamination in participants' strategies between environments. As we needed to counterbalance the presentation order of the three environments, the sample size had to be a multiple of 6 and thus simply we opted for 18 participants.

Data from an additional 26 participants were discarded before analysis based on pre-established criteria: the use of a wrong device (e.g. computer), insufficient score at the instructions quiz (less than six out of eight), a score not significantly higher than chance in the task, and the selection of the 'worst supplier' (the one presenting, after sampling, the smaller proportion of good-quality apricots) on 10% or more of the trials. We considered that such mistakes could occur when participants were confused and based their choice on the opposite colour or when they simply did not pay enough attention. As a result, three participants (in B32) were excluded because they used a wrong device, one participant (in W32) because s/he did not pass the instructions quiz, eight because of a score not significantly higher than chance (three in B10, one in W10, two in B32, and two in W32), 9 because of more than 10% 'worst choice' (four in B10, three in W10, and two in W32), and five because they did not satisfy the two last criteria (one in W10, four in B32).

## Analyses

Analyses were run using R and MATLAB. Normality of the data was tested using Shapiro tests and homoscedasticity was tested using F tests or Bartlett tests (for more than two samples). In cases where it was possible, parametric tests were preferred, otherwise non-parametric tests were used. One-sample Wilcoxon tests against the environment averaged outcome (25, 50, and 75 respectively for the poor, neutral, and rich environment) were used to test whether participant's final score was significantly different than chance.

## Sampling strategy

Our objective is to investigate how humans allocate limited resources to gather information about alternatives whose probability of success is unknown a priori. The resources are represented in our study by the number of samples available (sampling capacity, or coins) on a trial-by-trial basis. We first focus on how this limited capacity influences the number of alternatives participants chose to sample. To do so we introduce the notion of 'sampling strategy' characterized as the number of alternatives sampled ($M$) depending on the sampling capacity ($C$) available (*Figure 2*). The lower the ratios of $M/C$, the more the strategy tends to depth, whereas a ratio of 1 indicates pure breadth. We observe how closely empirical behaviour relates to the optimal strategy predicted theoretically (see *Moreno-Bote et al., 2020*) and whether it is affected by contextual parameters such as the overall probability of success (environment richness).

## Optimal sampling strategy

The optimal sampling strategy is modelled after *Moreno-Bote et al., 2020*, and we provide here the details. The assumptions of the framework are characterized by normative agents who don't show any memory leak and are aware of the environment priors ($\alpha$ and $\beta$). More precisely, normative agents are set to maximize expected reward and to select the sampled alternatives which maximize the normative value $V_i = \left( \sum_1^s O_{s,i} + \alpha \right) / \left( N_i + \alpha + \beta \right)$, where $O_{s,i}$ is the outcome of each sample $s$ (1 or 0) allocated in the alternative $i$, $N_i$ is the total number of samples allocated, and $\alpha$ and $\beta$ are the parameters that describe the beta distribution from where rewards in the environment are drawn. The normative strategy is described at two levels depending on both the resources available (capacity $C$) and the environment richness. Firstly, at the trial level, it predicts how many alternatives should be sampled (BD trade-off, *Appendix 1—figure 1A*). At low capacity (e.g. $C < 6$ for the poor environment), the optimal model predicts a pure breadth, meaning that each resource sample should be allocated in a different alternative. As capacity increases, we observe an abrupt change of regime with the optimal number of sampled alternatives being close to a power law of the capacity (*Moreno-Bote et al., 2020*). Intuitively, when the agent has more resources, it is best to focus samples on a few alternatives, rather than spreading them over too many, as the latter strategy will allow for very little discriminability between the quality of the sampled alternatives. The capacity at which the transition between pure breadth and BD trade-off happens directly depends on the richness of the environment, the poorer the environment is and the later (at higher capacity) the transition will occur. Secondly, the normative model predicts how many samples should be allocated in each of the sampled alternatives (*Appendix 1—figure 1B*). In deep allocations and rich environments especially, the optimal behaviour is defined by non-homogenous allocations of samples to break ties.

## Individual differences

We did not observe any effect of participants' gender on the BD trade-off (ANOVA: $F_1 = .21$, $p = .65$) nor an interaction between gender and environment richness effects (between: $F_1 = .31$, $p = .58$, within: $F_1 = .29$, $p = .60$). We did not observe either any effect of participants' age (median split: 18–23 vs. 24–52 years of age) on the BD trade-off ($F_1 = 1.45$, $p = .23$). Additionally, no significant interaction between age and environment richness was found (between: $F_1 = 3.20$, $p = .076$, within: $F_1 = .10$, $p = .78$).

## Effect of environment on sampling strategy

In order to characterize how participants' sampling strategies might differ depending on the richness of the environment, we decided to test three models to fit $M$ as a function of capacity $C$ separately to data from each participant and environment:

1. A piece-wise power-law model ($W$): $M(C) = \left\{ \begin{array}{l} C^{a_1} \; if \; C \leq B \\ C^{a_2} + b \; if \; C > B \end{array} \right\}$, where $B$ corresponds to the breakpoint with $B \in \{3, 4, \ldots, 9\}$ in narrow-capacity designs, and $B \in \{3, 4, \ldots, 16\}$ in wide-capacity designs.
2. A linear model ($L$): $M(C) = aC + b$.
3. A power-law model ($P$): $M(C) = C^a$.

Linear and power-law models were compared using a paired Wilcoxon test on individual R-squared adjusted while power-law and piece-wise power law models were compared using ANOVA (with $\alpha = .05$) at the participant and environment levels. The power-law model captured the empirical relationship between $C$ and $M$ best.

Given the results of the model comparisons, the effect of the environment on participants' sampling strategies was therefore tested using the power-law model. We compared the power factor $a$ extracted from power-law fits in each environment using within (designs W10 and W32) or between (designs B10 and B32) one-way ANOVA. Post hoc comparisons between environments were conducted using t-tests with Bonferroni correction for multiple comparisons. The four experimental designs were analysed individually, in order to account for the structure of the data (within- or between-subjects).

## Observed vs. optimal sampling strategy

Fitting $M$ as a function of capacity, we previously observed that even when participants' sampling strategy falls close to the predicted optimal one (see **Figure 2**), they do not coincide completely. To better characterize participants' deviations from optimality we first describe them overall, independently from the capacity, by fitting the optimal number of alternatives sampled ($M_{opt}$) as a function of capacity using the power-law model (described above) as it was the one better accounting for our data. We compared both power factors $a$ using one-sample t-tests in each environment individually, correcting for multiple comparisons, and observed significant differences, going in counter-directions depending on the environment. To further understand these deviations, we investigate if capacity could be a cause of modulations. To do so, we computed the differences between the number of alternatives observed $M$ and $M_{opt}$ for each capacity and environment (**Appendix 2—figure 1**) and tested for marginal effects of the capacity and the environment, as well as for an interaction between capacity and environment using a Sheirer-Ray-Hare test. All effects were found significant and post hoc analyses were performed using one-sample Wilcoxon tests (against the null hypothesis $mu = 0$, see **Supplementary file 1**) in order to better understand how both capacity and environment richness affect the nature and directions of participants' sampling deviations from optimality.

## Model comparison

The BD dilemma can be solved using extreme behaviours (pure-depth or pure-breadth) but also trade-offs which establish a balance between breadth and depth and often lead to more optimal choices. In the current study, we quantitatively assess which strategy describes best the individual participants' sampling behaviour by comparing the ability of the optimal and heuristic models (characterizing both extreme and trade-off behaviours) to predict the data. The number of vendors $M$ that participant $k$ chose in trial $j$, denoted $M_j^k$, was predicted by using one of three models, corrupted by either Gaussian or binomial noise. Each model predicts the mean $M\left(C_j^k\right)$ number of selected vendors given that capacity was $C_j^k$. We use the following models:

1. *Optimal* model: $M\left(C\right)$ follows the ideal observer $M_{opt}\left(C\right)$

   $M\left(C\right) = M_{opt}\left(C\right)$.

   The ideal observer corresponds to the sample allocation that would maximize expected reward for each capacity in each environment. It describes the optimal number of suppliers to be sampled *M*, as well as the distribution of samples per supplier. The ideal observer was described in a previous theoretical paper (**Moreno-Bote et al., 2020**) and here has been calculated with Monte-Carlo simulations.

2. *Linear models*: We used three variants of linear model:
   a. *Depth* model: $M\left(C\right)$ is *constant* and independent of capacity $C$, with $M$ following an almost pure-depth strategy,

   $M\left(C\right) = 2$.
   b. *Pure-breadth* model: always equals to ,

   $M\left(C\right) = C$.
   c. *Free-slope linear* model: $M\left(C\right)$ increases with capacity $C$ with a free factor $d$,

   $M\left(C\right) = dC$.

3. *Power-law* model: We used two variants of this model:
   a. *Square root* model: $M\left(C\right)$ increases with the square root of capacity $C$.

$$M(C) = \sqrt{C}.$$

b. *Free power-law* model: $M(C)$ increases with capacity $C$ to the $w$ th power,

$$M(C) = C^w.$$

For each model, to compute the likelihood of the data $M_j^k$ for each trial $j$ and participant $k$, we assume a noise model that is either Gaussian or binomial. In the Gaussian noise model, we assume that $M_j^k = M\left(C_j^k\right) + \sigma_j^k z_j^k$, where $z_j^k$ is independently and identically distributed standard normal noise, and $\sigma_j^k = a^k + b^k C_j^k$. In all cases we obtained that the best fit parameters obeyed $a^k \neq 0$, $b^k \neq 0$, so that $\sigma_j^k \neq 0$ *and* $\left(\sigma_j^k\right)^2 > 0$ for all values of $C$. The per-participant likelihood of the model is the sum of individual trial-by-trial likelihoods

$$\mathfrak{L}_N\left(a^k, b^k\right) = \sum_j \left( -\log\left(\sqrt{2\pi}\right) - \log\left(\sigma_j^{k2}\right) - \frac{\left(M_j^k - M\left(C_j^k\right) - \mu^k\right)^2}{2\sigma_j^{k2}} \right).$$

In the binomial case, we assume that the observed $M_j^k$ follows a binomial distribution $M_j^k \sim B\left(n^k, p^k\right)$ with $p^k \in [0, 1]$ and

$$n^k = \frac{M\left(C_j^k\right) + \mu^k}{p^k}, \quad n^k = 1, 2, \ldots, N \text{ and } n^k \geq M_j^k$$

so that the expectation of $M$ equals the desired model's expectation $M(C)$. In this case, the per-participant likelihood of the data under the model and binomial noise is

$$\mathfrak{L}_B\left(p^k\right) = \sum_j \left( \log\left(\frac{n^k!}{M_j^k!\left(n^k - M_j^k\right)!}\right) + M_j^k \log\left(p^k\right) + \left(n^k - M_j^k\right)\log\left(1 - p^k\right) \right)$$

The overall log likelihood of each model was the sum of log likelihoods across trials, participants, and environments.

Each model (1–3 above) with each noise model (Gaussian or binomial) was fit and tested to each participant and environment (in the case of within-subjects designs) individually using fourfold cross-validation. Specifically, in onefold, 75% of the data in each participant was used to fit the per-participant model parameters by maximizing their individual log likelihoods. The remaining 25% of the data was used to measure the quality of the fit. This was quantified as the log likelihood of the model with its fitted parameters in the test set, and summed across participants, and averaged over the fourfolds (four different ways of taking 25% non-overlapping trials for testing and the remaining for training). We call the resulting average CVLL. In the binomial noise model, the log likelihood of the test set becomes minus infinity in trials when $n < M$, which is an impossible event. This concerned a small set of trials in 27 participants in the square root model, 31 in the optimal model, 16 in the depth model, 69 in the free power-law model, and 48 in the linear model. However, in all cases it affected only one out of the fourfolds, so the averaged log likelihood could be computed across the three remaining folds. Model comparison was performed by comparing the CVLL across models and noise models using Wilcoxon matched pairs signed-ranks tests with Bonferroni correction. In order to ascertain the results of the log likelihood-based model comparisons, the models were also compared using AIC (*Akaike, 1977*), which were computed on the whole dataset. Both methods provide qualitatively identical results (compare *Figure 8* with *Figure 8—figure supplement 3* for binomial noise models and *Figure 8—figure supplement 1* with *Figure 8—figure supplement 4* for Gaussian noise models). Free parameter $a$ was optimized on a grid from –3 to –1 and $b$ on a grid from –1 to –1, both with a step of 0.05 in the Gaussian noise model and the parameter $p$ in the binomial noise model which was optimized on grid from 0 to 1 with a step of 0.01. The free parameter $w$ was optimized on a grid from 0.05 to 1.1 with a step of 0.05 in both Gaussian and binomial noise models.

## Exploring how samples are dispatched within the alternatives sampled (splits)

To descript more precisely participants' sampling strategy, we studied how the samples are allocated inside the selected suppliers. Indeed, for an identical number of alternatives sampled $M$ and capacity

$C$, we can observe various allocations of samples. For example, $M = 2$ and $C = 4$ can result in the allocation of 2 samples in two different alternatives each; {2,2}, or in the allocation of 3 samples in a first alternative and 1 sample in a second; {3,1}. Visually, we observe that participants have a tendency to allocate the samples homogeneously across the sampled alternatives (e.g. {2,2}) (see *Figure 9A*). In order to statistically capture this putative bias, we computed the standard deviations of the sample allocations. An homogenous allocation of samples would result in a standard deviations close to 0 (e.g. s.d.({2,2})=0) whereas a more heterogeneous allocation of samples would be associated with higher standard deviations (e.g. s.d.({1,3}) ≈ 1.41). We computed the standard deviation of each sample allocation and compared it to the standard deviation of the optimal allocation of samples (predicted by *Moreno-Bote et al., 2020*) in each environment using Wilcoxon tests with Bonferroni corrections to adjust for the multiple comparisons. We tested the effect of the environment on the magnitude of these differences using a Kruskal-Wallis test and performed post hoc analyses using Wilcoxon tests with Bonferroni corrections.

We also computed the differences between the outcomes observed in our data with the ones obtained when following the ideal observer strategy. A Kruskal test was performed to test the significant effect of the environment on the outcome differences and one-sample Wilcoxon tests with Bonferroni corrections were then used to assess whether observed outcomes were significantly different than optimal outcomes inside each environment condition.

These analyses were performed for all trials (from all experimental designs) where the optimal number of alternatives sampled was strictly inferior to the capacity ($M_{opt} < C$) (see *Figure 9B–C*). This condition has been introduced because when $M_{opt} = C$, the optimal sample allocation follows a pure-breadth strategy, and its associated standard deviation is 0. Therefore, the standard deviation of the sample allocation observed in the data can only be superior to the optimal sample allocation which impairs the observation of participants' bias to sample homogenously the alternatives.

## Acknowledgements

This work is supported by the Howard Hughes Medical Institute (HHMI, Ref: 55008742), MINECO (Spain; BFU2017-85936-P), ICREA Academia (2016) and Ministerio de Ciencia e Innovación (Ref: PID2020-114196GB-I00/AEI) to RM-B. SS-. is funded by Ministerio de Ciencia e Innovación (Ref: PID2019-108531GB-I00 AEI/FEDER) and the FEDER/ERDF Operative Programme for Catalunya 2014-2020. AV is supported by a FI fellowship from the AGAUR (2019FI_B 00302). We would like to thank Pierre Marais for his precious help in setting up the Flutter application and coding the experimental task.

## Additional information

### Funding

| Funder | Grant reference number | Author |
| --- | --- | --- |
| Howard Hughes Medical Institute | 55008742 | Rubén Moreno-Bote |
| Institució Catalana de Recerca i Estudis Avançats | 2016 | Rubén Moreno-Bote |
| Ministerio de Ciencia e Innovación | PID2019-108531GB-I00 AEI/FEDER | Salvador Soto-Faraco |
| European Regional Development Fund | Operative Programme for Catalunya 2014-2020 | Salvador Soto-Faraco |
| Agència de Gestió d'Ajuts Universitaris i de Recerca | 2019FI_B 00302 | Alice Vidal |

The funders had no role in study design, data collection and interpretation, or the decision to submit the work for publication.

## Author contributions
Alice Vidal, Conceptualization, Data curation, Software, Formal analysis, Investigation, Visualization, Methodology, Writing - original draft, Writing – review and editing; Salvador Soto-Faraco, Rubén Moreno-Bote, Conceptualization, Supervision, Funding acquisition, Methodology, Writing – review and editing

## Author ORCIDs
Alice Vidal ⓘ http://orcid.org/0000-0003-4477-510X
Salvador Soto-Faraco ⓘ http://orcid.org/0000-0002-4799-3762

## Ethics
Human subjects: Before starting the experiment, participants had to give their informed consent. This study was part of the project 'IMC: INTEGRACIÓN MULTISENSORIAL Y CONFLICTO' (PID2019-108531GB-I00) for which an ethical approval was obtained.

## Decision letter and Author response
Decision letter https://doi.org/10.7554/eLife.76985.sa1
Author response https://doi.org/10.7554/eLife.76985.sa2

---

# Additional files

## Supplementary files
• Transparent reporting form

• Supplementary file 1. Participants' sampling strategy deviates from optimality and tends to be tilted toward depth at low capacity and breadth at high capacity.

• Supplementary file 2. Post-hoc comparisons between deviations from optimality in power factors extracted from fitting the power law model to BD trade-offs in first and second halves of blocks separately.

• Supplementary file 3. Post-hoc analyses of the deviations from optimality in power factors extracted from fitting the power law model to BD trade-offs in first and second halves of blocks separately.

• Supplementary file 4. Post-hoc comparisons between deviations from optimality in power factors extracted from fitting the power law model to BD trade-offs in first and second halves of blocks separately.

• Supplementary file 5. Summary of the pair-wise comparisons of the 4-folds averaged CVLL between all six models using Gaussian distributed noise.

• Supplementary file 6. Summary of the comparisons between the averaged individual 4-fold CVLL in each environment and experimental design using Gaussian distributed noise.

• Supplementary file 7. Summary of the pair-wise comparisons of the individual AIC between all six models using Binomial distributed noise.

• Supplementary file 8. Summary of the comparisons between the individual AIC in each environment and experimental design using Binomial distributed noise.

• Supplementary file 9. Summary of the pair-wise comparisons of the individual AIC between all six models using Gaussian distributed noise.

• Supplementary file 10. Summary of the comparisons between the individual AIC in each environment and experimental design using Gaussian distributed noise.

## Data availability
The data and analysis scripts have been deposited in an OSF repository available here https://osf.io/kdbqs/?view_only=386d3bde49394e6bb88d247adc52b9ad.

The following dataset was generated:

| Author(s) | Year | Dataset title | Dataset URL | Database and Identifier |
|---|---|---|---|---|
| Vidal A, Soto-Faraco S, Moreno-Bote R | 2021 | Humans balance breadth and depth: Near-optimal performance in many-alternative decision making | https://osf.io/kdbqs/?view_only=386d3bde49394e6bb88d247adc52b9ad | Open Science Framework, kdbqs |

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

# Appendix 1

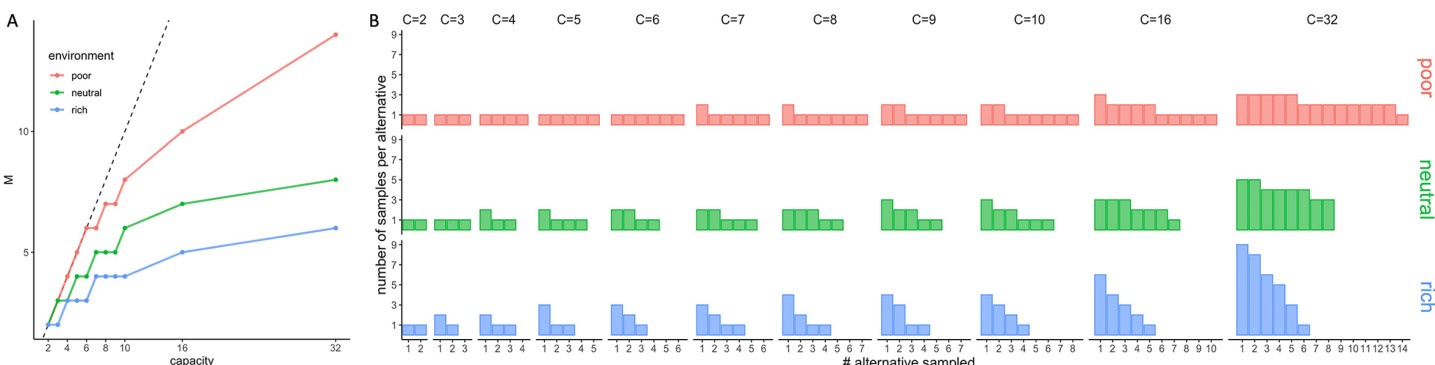

**Appendix 1—figure 1.** Optimal strategies as a function of both capacity and environment richness. (**A**) Optimal number of alternatives sampled $M$ as a function of the capacity for each of the three environments (colours). Dashed line indicates unit slope line (pure breadth). (**B**) Optimal number of samples allocated to each sampled alternative depending on the sampling capacity $C$ and the environment richness.

## Appendix 2

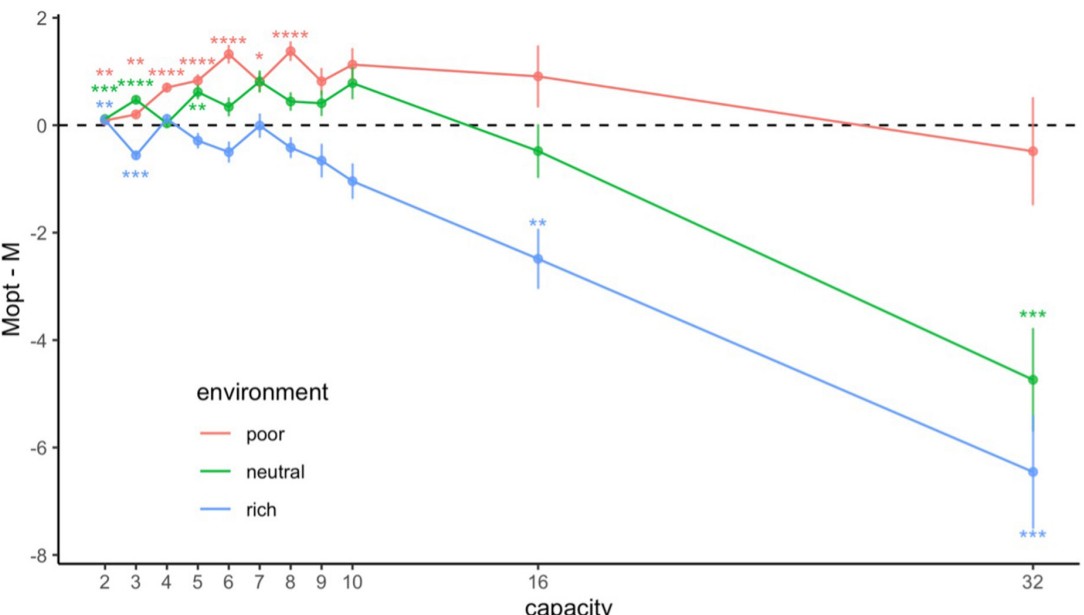

**Appendix 2—figure 1.** Participants' sampling strategy deviates from optimality and tends to be tilted toward depth at low capacity and breadth at high capacity. Difference between the optimal ($M_{opt}$) and the observed ($M$) number of alternatives sampled averaged across all participants for each capacity and environment (coloured lines). Error bars represent the standard error of the mean and significant deviations from 0 (dashed line) are marked as follows: '*': $p_{adj} < .05$, '**': $p_{adj} < .01$, '***': $p_{adj} < .001$, '****': $p_{adj} < .0001$.

## Appendix 3

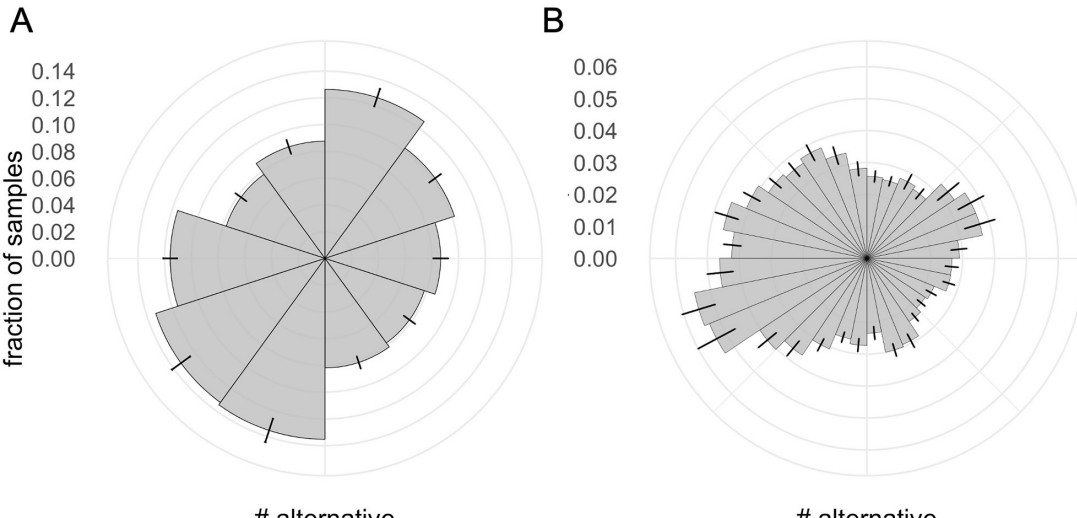

**Appendix 3—figure 1.** Participants present a motor bias when sampling. (**A–B**) Fraction of samples allocated in each alternative, in the designs with 10 alternatives (**W10 and B10, A**) and 32 alternatives (**W32 and B32, B**). Bars represent s.e.m.

## Appendix 4

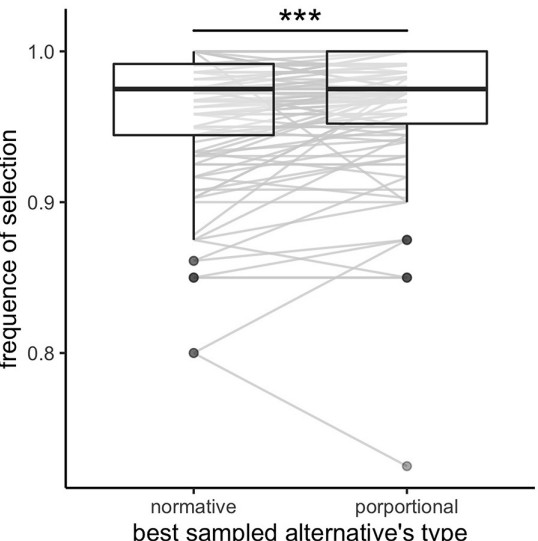

**Appendix 4—figure 1.** Participants select more often the best sampled alternative independently, compared to dependently, on the priors of the environment. Frequency of selection of the best sampled alternative calculated based on the normative outcome (depends on the environment richness) and the proportional outcome. Grey lines connect individual data. Lower and upper hinges correspond to the 1st and 3rd quartiles and vertical black lines represent IQR*1.5. `***`: p<0.001. Sample sizes per condition: n=126.

