## [Editor Report]

The authors describe human behavior in a novel task to understand how humans seek information about uncertain options when having a limited sampling budget – the 'breadth-depth' trade-off. They show that human information search approximates the optimal allocation strategy, but deviates from it by favoring breadth in poor environments and depth in rich environments. This study will likely be of interest to a broad range of behavioral and cognitive neuroscientists.

---

## [Decision Letter]

**Decision letter after peer review:**

Thank you for submitting your article "Balance between breadth and depth in human many-alternative decisions" for consideration by *eLife*. Your article has been reviewed by 2 peer reviewers, and the evaluation has been overseen by Valentin Wyart as the Reviewing Editor and Michael Frank as the Senior Editor. The following individual involved in the review of your submission has agreed to reveal their identity: Konstantinos Tsetsos (Reviewer #1).

The reviewers have discussed their reviews with one another, and the Reviewing Editor has drafted this to help you prepare a revised submission. We are very sorry for the delay in getting back to you with a decision. Both reviewers have found that your work addresses a timely and ecologically valid, yet rarely studied question. Your task and analyses offer an elegant approach to modulating the breadth-depth trade-off using a variety of contextual factors. However, the reviewers have also identified different points that should be addressed before considering the manuscript suitable for publication in *eLife*. After discussion with the reviewers, we have prepared a list of essential revisions (listed below) that should be carried out and detailed in a point-by-point response letter. These essential revisions point toward the possible collection of additional data, but this is not required as long as the limitations of the currently reported data are clearly mentioned in the Discussion section of the manuscript. We hope that you will find these reviews helpful when revising your manuscript and that you will be able to address these different concerns. The individual reviews from the two reviewers are copied at the bottom of this message for your information regarding additional concerns, but they do not require point-by-point responses.

Essential Revisions:

1) Normative framework. Both reviewers have noted that 'optimal resource allocation' framework appears central to the study, but is not adequately presented in the manuscript. The following questions should be answered for readers in the revised manuscript: What are the assumptions in the normative framework? (e.g., are normative agents aware of the statistics of the environment?) What is the objective/cost function used by normative agents? It would also be particularly helpful to find intuitions in the revised manuscript about how the normative strategy changes as a function of the various contextual factors.

2) Computational modeling. The computational model used to describe participants' behavior is largely descriptive, in the sense that it focuses on summary statistics (e.g., the power factor) that are characteristic of the normative strategy. This descriptive modeling has clear merits, but it is difficult to understand from it the cognitive mechanisms that underlie breadth-depth behavior and their changes across different contextual factors. A more complete/detailed characterization of human behavior in the task is needed, along the following axes at least. Does search behavior change as a function of the reward accumulated (see also point 4)? Do participants show 'choice-history' biases – placing coins on the same alternatives across trials or on alternatives that were previously rewarding (see also point 3)? In 'deep' allocation trials, do they allocate coins by alternating across alternatives, or do they place coins all at once on a certain alternative? Finally, do they make correct decisions by choosing optimally the alternative based on the revealed information (see also point 3)?

3) Origin of sequential effects. The presence of sequential effects in participants' behavior is interesting. However, it is not clear whether these effects reflect genuine 'choice-history' biases or just the fact that participants have no information about changes in the statistics of the environment and thus carry over the previous strategy until they notice these changes. And more generally, it is unclear whether the observed behavioral patterns represent ecologically valid behavior and merely an idiosyncratic solution to the laboratory task developed by the authors. There are different ways through which the authors could address these concerns. First, by examining sequential biases dynamically within a block, and seeing how and when they dissipate. Second, by studying whether homogenous sampling corresponds to a global systematic bias or rather the result of understanding the statistics of each option? Third, does the reward obtained in the previous trial influence allocation in the succeeding trial, over and above the environment statistics? Last, it would be very useful to discuss in the revised manuscript whether the biases observed in the task may translate into more general information sampling behavior. For example, whether the block-wise manipulation used in the task represents an ecologically valid decision-making problem could be discussed.

4) Relation to the explore-exploit trade-off. It would be very useful to use other aspects of the task to study how participants' behavior in this breadth-depth dilemma is influenced by the widely studied explore-exploit trade-off. First, it is important to know how participants decide after the samples have been revealed. Do participants always choose the optimal alternative? How do they account for uncertainty and ambiguity? For example, do they tend to prefer the alternative from which they sampled more, even if another alternative has a larger ratio of positive/negative samples? Or, when all sampled alternatives appear poor, do they occasionally bet on an unsampled alternative? The suggestion that exploration-exploitation occurs in parallel to the breadth-depth dilemma should be compared to the possibility that the two interact with each other. During exploration, behavior would be directed towards more uncertain (less sampled) options. By contrast, during exploitation, behavior is directed towards options that are most likely to be rewarding. Here, strategies that focus on depth could overlap with exploitative behavior, whereas strategies that sample superficially from many options could overlap with exploratory behavior.

5) Individual differences. It is currently unclear whether there are individual differences across these breadth-depth sampling strategies and how these differences could be interpreted in light of their associated cognitive mechanisms. Are there individual differences that can be related to differential use of strategies or transition between strategies across contextual factors? What implications do these differences have for understanding the breadth-depth dilemma? Importantly, knowing whether the behavior described in the task relates to real-life information sampling behavior would have largely benefitted from correlations with psychological questionnaires targeting these cognitive traits. However, this would require the collection of additional data. Instead of collecting these additional data, the authors should include a dedicated paragraph in the Discussion section to mention this point, and point toward additional work testing this relation of task vs. real-life behavior in future work.

*Reviewer #1 (Recommendations for the authors):*

This paper provides a comprehensive experimental exploration of the breadth vs. depth trade-off. The authors, building off previous theoretical work, introduce a novel experimental task and examine how a set of factors- such as the available search resources, the number of alternatives, and the average expected value of the alternatives (i.e., how rich the environment is) – influence information search. The behavioural results indicate that people's search strategy reflects the optimal policy, with increased resources and richer environments promoting depth over breadth. Despite the fact that the qualitative predictions of the optimal allocation framework are roughly met, human behaviour exhibits some discrepancies from the optimal policy.

Strengths

1) The paper addresses a timely and ecologically valid, yet overlooked question. The experimental work presented here builds off a solid theoretical framework.

2) The experimental data are rich enough and the quantitative analysis employed is rigorous.

3) There is a fair discussion in relation to the optimal policy, with the authors describing accurately the ways in which the data are consistent and inconsistent with normative predictions.

Weaknesses

1) The optimal resources allocation framework is central to this paper but yet not adequately described. What are the assumptions in the normative framework (are normative agents aware of the environmental statistics) and what is the objective/ cost function? The framework needs to be introduced more comprehensively. Additionally, intuitions about how the optimal strategy changes as a function of various factors would be helpful.

2) The computational modelling is largely descriptive, in the sense that it focuses on a few quantities (e.g., power factor) that are related to the optimal policy. What is currently missing is a more complete characterisation of human behaviour in the task. Does search behaviour change as a function of reward accumulated? Do participants show serial choice biases (placing coins on the same alternatives across trials or on alternatives that were previously rewarding)? In more "deep" allocation trials, do they allocate coins by alternating across alternatives, or do they place coins all at once on a certain alternative? Finally, do they make correct decisions by choosing optimally the alternative based on the revealed information?

3) It is not clear if the sequential effects reflect genuine biases or just the fact that participants have no information about changes in the statistics of the environment and thus carry over the previous strategy until realising that the environment has changed. Examining these biases dynamically, within a block, and seeing how and when they dissipated is relevant.

Recommendations

1) It would be interesting to show how participants decide after the samples have been revealed (and as mentioned in the "weaknesses" part, whether decisions show serial choice biases etc). Do participants always choose the optimal alternative? How do they deal with uncertainty and ambiguity? For example, do they tend to prefer the alternative from which they sampled more, even if another alternative has a larger ratio of positive/ negative samples? Or, when all sampled alternatives appear poor, do they perhaps "bet" on an unsampled alternative?

2) Does the reward obtained in the previous trial influence allocation in the succeeding trial, over and above the environment statistics?

3) The paper is well written but would benefit from proofreading and a bit less technical description of the results.

4) It would be good to provide intuitions on the optimal policy around Figure 5 and homogenous allocation.

5) Unlike participants, I believe that the optimal model operates on full knowledge of the environment. What are the dynamics of allocation within each block? Do participants take time to adjust and do they adjust their policy at all? Examining trial-by-trial model parameters (e.g., using a leave-one-out approach) might be informative.

*Reviewer #2 (Recommendations for the authors):*

Vidal and colleagues aimed to understand information sampling under finite resources. They suggest two potential manners that can be used to sample information: sampling a lot of information from a few options (depth) or sampling less information from many options (breadth). Results show that behavioural patterns can be composed of either a pure breadth or depth strategy, while other times behavioural patterns rely on a mixture of both. In general, the main factors driving this dissociation are the number of options one can sample from (many options require more breadth than depth) and the richness of the environment (richer environments allow for more depth).

The question of sampling information under constrained resources is relevant and interesting. The task allows to test this question and offers an elegant tool to modulate the information sampling process by a variety of contextual factors. Even though the task shares similarities with classic bandit tasks to test exploration-exploitation dilemma, here the critical difference is the manipulation of certain contextual factors in a block-wise structure. On the one hand, it is interesting because it might allow testing in more detail what occurs within phases of exploration/exploitation (e.g. depth ~ exploitation, breadth ~ exploration), while on the other hand, it raises the question of whether a block-wise manipulation represents realistic decision-making problems.

One of the weaker points is to understand the mechanisms that underlie breadth-depth behaviour and their transition. Even though the authors use a power law model to simulate behaviour, it is unclear what this exactly means for cognitive processes that might change depending on the different strategies. Further, it is unclear whether there are individual differences across these strategies and how these could be interpreted given the potential cognitive mechanisms of each strategy.

This relates to a second weakness which is the differentiation of whether behavioural patterns are simply a solution to the current task or whether they represent realistic behaviour. For example, is homogenous sampling a more global systematic bias or is it the result of understanding the underlying probability distribution of each option? In other words, it might be easier to integrate observations from an equal number of samples to evaluate the quality of each option, however it is questionable whether this bias translates into more general information sampling behaviour. It would have been interesting to extend and thereby validate the importance of the results by (a) relating them to individual differences, (b) testing differences in strategies and their transitions during interleaved vs. block-wise trials, and potentially (c) relating them to psychological questionnaires to understand the relevance of the results.

1. The authors have shown that the order of blocks might impact the strategy in the next block. Can the authors speculate on whether they think interleaved trials rather than blocks would show different results? If so, how relevant is the current task in capturing realistic information sampling under constraints?

2. Are there individual differences that can be related to different use of strategies or transition between strategies? What implications do these differences have?

---

## [Author Response]

Essential Revisions:1) Normative framework. Both reviewers have noted that 'optimal resource allocation' framework appears central to the study, but is not adequately presented in the manuscript. The following questions should be answered for readers in the revised manuscript: What are the assumptions in the normative framework? (e.g., are normative agents aware of the statistics of the environment?) What is the objective/cost function used by normative agents? It would also be particularly helpful to find intuitions in the revised manuscript about how the normative strategy changes as a function of the various contextual factors.

Thanks for a lot for the suggestions and questions. We agree that providing more information about the model and its assumptions will help to improve the clarity of the paper. Therefore, following the editor and reviewers’ recommendations, we have added a paragraph in the Methods section (“optimal sampling strategy” p.27) to better describe the model and its assumptions, accompanied with supporting figures describing the evolution of the normative models as a function of capacity and environment richness (Appendix 1 – Figure 1, p.56). See further details in the paragraph *Experimental design* of the Methods section (p.24-25).

2) Computational modeling. The computational model used to describe participants' behavior is largely descriptive, in the sense that it focuses on summary statistics (e.g., the power factor) that are characteristic of the normative strategy. This descriptive modeling has clear merits, but it is difficult to understand from it the cognitive mechanisms that underlie breadth-depth behavior and their changes across different contextual factors. A more complete/detailed characterization of human behavior in the task is needed, along the following axes at least. Does search behavior change as a function of the reward accumulated (see also point 4)?

Thanks a lot for the question. In the experiment, the amount of reward accumulated inside a particular environment (block) directly depends on time and therefore on the participants experience in the task. Hence, analysing the BD trade-off depending on the reward accumulated throughout the block is almost equivalent to analysing it as a function of time. We find that there is a learning effect within a block, whereby in the second half of the block participant are closer to optimal than in the first part. We have reported this leaning effect in Section ‘deviations from optimality’, fourth paragraph, p.9 and Figure 4 (note: we have found almost identical effects when redoing the analysis by splitting accumulated reward; not shown).

In the above response, we have assumed the reviewer’s point related to longer time constants (accumulated effects). However, whilst discussing how to address this point, we also considered the possibility that the reviewers’ suggestion referred to another (also interesting) question, which is the short-term effects of reward on the BD tradeoffs. To address this second aspect of the question, we run additional tests to investigate whether reward has a short-term effect on participants sampling strategy, reported in the Section ‘short-term sequential effects are absent or weak’, first paragraph, p.12 and Figure 6. In summary, we demonstrated that the magnitude of reward obtained in one trial does not seem to directly affect the BD trade-off on the consecutive trial.

Do participants show 'choice-history' biases – placing coins on the same alternatives across trials or on alternatives that were previously rewarding (see also point 3)?

Thanks a lot for the question. To investigate whether alternatives which were previously rewarded influence future sampling, we split trials depending on the magnitude of the reward (high or low) obtained in the previous trial (median split, corrected for capacity). We observed a tendency for participants to resample (allocate at least one sample) more often the previously chosen alternative after high reward (see Section ‘short-term sequential effects are absent or weak’, second paragraph, p.13 and Figure 7A). A similar bias is observed when considering, at trial t+1, the fraction of samples allocated in the previously selected alternative, at trial t (Figure 7B). Crucially, we observed that the presence of this sequential effect at the individual level did not affect sampling strategies (Figure 7C). We also observed that alternatives are not equally sampled as a result of possible motor biases (see Appendix 3 – Figure 1, p.58), but due to the assignment of the random rewards throughout the experiment, such bias was easily taken into account to study the more relevant biases.

In 'deep' allocation trials, do they allocate coins by alternating across alternatives, or do they place coins all at once on a certain alternative?

In the previous version of the paper, we report that on 39.9% of the trials, participants first allocate all the samples to one alternative before sampling another one (strategy focused on the number of samples per alternatives, so on *depth*), while on 21.8% of the trials they chose to first sample once each chosen alternative M (strategy focused on M, so on *breadth*) (see p.18-19). Therefore, people tend to use a mix strategy. Prompted by the reviewers’ question, we now went a step further by analyzing how these fractions depend on the capacity C and the number M of alternatives sampled. Overall, people tend to use more a *depth* focused strategy than a *breadth* focused one, independently of the depth of the sample allocation (M/C ratio). In contrast, and as one would expect, the strategy focused on *breadth* prevails only for trials with a rather shallow allocation of samples (M close to C).

We report these additional analyses in the last paragraph of the Results section

‘participants tend to sample homogeneously amongst alternatives’ (p.19) and in Figure

10.

Finally, do they make correct decisions by choosing optimally the alternative based on the revealed information (see also point 3)?

A detailed answer to this question is provided bellow in point 4.a.

3) Origin of sequential effects. The presence of sequential effects in participants' behavior is interesting. However, it is not clear whether these effects reflect genuine 'choice-history' biases or just the fact that participants have no information about changes in the statistics of the environment and thus carry over the previous strategy until they notice these changes. And more generally, it is unclear whether the observed behavioral patterns represent ecologically valid behavior and merely an idiosyncratic solution to the laboratory task developed by the authors. There are different ways through which the authors could address these concerns. First, by examining sequential biases dynamically within a block, and seeing how and when they dissipate.

Thanks for the question. First, we’d like to point out that participants are told when the environment changes between different blocks, and about the characteristics of the new environment (e.g., ‘a majority of the suppliers have a low/average/high proportion of good quality apricots’, we have clarified this point in the Methods, p.25). Further, they are also presented with 10 practice trials to get acquainted with the new environment. This said, the sequential bias observed could still be due to carryover effects from the previously used strategy.

In blocks following an environment change compared to those presented first or alone (that is without environment change), we observe a significant interaction between experience in block (first or second block’s half) and the environment showing that participants seem to be closer to optimal in the second compared to the first half of the block (post-hoc comparisons are reported in a table in the Supplementary File 2, p.46 and in Figure 5 —figure supplement 2A, p.38). Additionally, the magnitude of this improvement is not statistically different depending on the block’s history (with or without environment change). However, the contamination effects observed in blocks presented after another environment (environment change) do not seem to dissipate as a function of experience (time) within a block, because the block's history (presented first or after a poor, neutral or rich environment) still has a significant effect on the power factor a deviation from optimality (a_observed_− a_optimal_) in the second half of the block (post-hoc comparisons are reported in a table in Supplementary File 3, p.47 and in Figure 5 —figure supplement 2B, p.38). Therefore, the carry over effects from the previously used strategy are long-lived. We do not currently know the origin of these carryover effects, as we showed above that the amount of reward does not have a short-term effect on the BD dilemma (see ‘choice history biases’ in Results section).

These analyses were added in the Results section (‘deviations from optimality’, p.1112) We also added a simpler analysis to test sequential effects due to the environment change using the optimal BD trade-off as a baseline instead of using the first environment presented, as the previous analysis was indeed noisier due to our limited sample size. We added a Figure (Figure 5) describing the results.

Second, by studying whether homogenous sampling corresponds to a global systematic bias or rather the result of understanding the statistics of each option?

In our opinion, it is difficult to detangle the origin of the homogenous sampling with our current setting. However, we can say that participants have a good understanding of the statistics of the sampled options, as we observed (see point 4.a bellow) that on a great majority of trials (more than 96%) participants select, as a final choice the option which shows the best success probability among the ones sampled (we mention this argument in the discussion p.22). One speculation is that using a homogenous sampling is a cognitive shortcut. That is, participants may sample intentionally homogenously to facilitate the comparisons between the sampled alternatives. We have elaborated more on this potential account in the manuscript (discussion, p.22).

Third, does the reward obtained in the previous trial influence allocation in the succeeding trial, over and above the environment statistics?

We observe that the reward obtained in the previous trial influence participants’ probability to resample the alternative selected for the final choice, but we demonstrated that this bias did not seem to influence the fact that the BD trade-off depends on the environment richness (see point 2.b). We also investigated whether the BD trade-off is modulated by the magnitude of reward obtained in the previous trial and did not find any significant effect (see point 2.a).

Last, it would be very useful to discuss in the revised manuscript whether the biases observed in the task may translate into more general information sampling behavior.

Thanks a lot for the suggestion. We have added text in the discussion (p.22) to further discuss the general validity of the homogeneous sampling bias described in our work and also biases related to participants’ tendency to sample more deeply than optimal, especially in poorer environments and at small capacity (see Appendix 2 – Figure 1, p.57).

In relation to the homogeneous sampling bias, we speculate that homogenous sampling may simplify the choice in two ways: by facilitating the comparisons of success probabilities of the sampled alternatives (as the proportional outcome, described in point 4a, would be similar or different), and by removing the possibility that one choice is riskier than another as each alternative carries the same amount of information. We mention that this last point may be driven by humans’ preference to sample the most uncertain option (‘information-directed exploration’). In relation to the deep bias, it could originate a tendency to reduce uncertainty about the outcome of the sampled alternative, and therefore somehow connected to the homogenous sampling bias. However, whether any of these biases are related to other biases, such as risk-aversion, remain to be explored.

For example, whether the block-wise manipulation used in the task represents an ecologically valid decision-making problem could be discussed.

In some natural situations, for example when navigating online and switching from one website to another, the environment can rapidly change. However, it is equally true that in a good range of everyday situations where environment characteristics tend to remain stable. For example, when deciding in real life, the quality or richness of the environment often depends on one’s location (a neighborhood, a store, a city), and therefore it is unlikely to change abruptly. Our experimental design, with environment richness manipulated in a block-wise manner, provides a good model for this type of situations, whilst limiting the cognitive cost used for adapting to a new environment.

4) Relation to the explore-exploit trade-off. It would be very useful to use other aspects of the task to study how participants' behavior in this breadth-depth dilemma is influenced by the widely studied explore-exploit trade-off. First, it is important to know how participants decide after the samples have been revealed. Do participants always choose the optimal alternative? How do they account for uncertainty and ambiguity? For example, do they tend to prefer the alternative from which they sampled more, even if another alternative has a larger ratio of positive/negative samples?

In the normative model the optimal sampled alternative is the alternative i that maximizes the normative value Vnormi=(∑Os,i+α)Ni+α+β where α and β are the parameters describing the prior distribution of rewards in the current environment. We observed that participants select the optimal sampled alternative on 96.0±4.34% (mean±sd) of the trial. We also observed that for 90% of the participants, this proportion is superior to 90.3%. We report this and other related results in Section ‘Optimal or close-to-optimal sampled alternative are mostly chosen’, p.19-20 and Appendix 4 – Figure 1, p.59 (including your correct intuition about a bias for larger sample size, etc.). These results clearly show that participants understand the task and that they mostly select the optimal or close-to-optimal sampled alternative.

Or, when all sampled alternatives appear poor, do they occasionally bet on an unsampled alternative?

In the experiment, participants did not have the possibility to select an unsampled alternative. We apologise if this wasn’t clear enough and we added a sentence to emphasise this experimental constraint (Results, p.4 and Methods, p.25).

The suggestion that exploration-exploitation occurs in parallel to the breadth-depth dilemma should be compared to the possibility that the two interact with each other. During exploration, behavior would be directed towards more uncertain (less sampled) options. By contrast, during exploitation, behavior is directed towards options that are most likely to be rewarding. Here, strategies that focus on depth could overlap with exploitative behavior, whereas strategies that sample superficially from many options could overlap with exploratory behavior.

Thanks for the suggestion. We have now acknowledged this possibility in the first paragraph of the Discussion (p.20). In our opinion, EE and BD will strongly interact in real world conditions, so we largely agree with the reviewers, which is a matter of needed, future work.

5) Individual differences. It is currently unclear whether there are individual differences across these breadth-depth sampling strategies and how these differences could be interpreted in light of their associated cognitive mechanisms. Are there individual differences that can be related to differential use of strategies or transition between strategies across contextual factors? What implications do these differences have for understanding the breadth-depth dilemma? Importantly, knowing whether the behavior described in the task relates to real-life information sampling behavior would have largely benefitted from correlations with psychological questionnaires targeting these cognitive traits. However, this would require the collection of additional data. Instead of collecting these additional data, the authors should include a dedicated paragraph in the Discussion section to mention this point, and point toward additional work testing this relation of task vs. real-life behavior in future work.

Thanks for the questions. We did notice in our data set, individual differences in the way participants adapt their strategy to both the resources available (capacity) and the environment richness and how these strategies relate to optimality (in Figure 2A are displayed individual BD trade-offs).

First, we reported in Methods, in the section ‘individual differences’ (p.27), the absence of an effect of gender and age on the breadth-depth trade-off. However, our sample is not optimal to test those potential effects (little power to detect a small effect size and age not uniformly distributed). In the future, the BD apricot task could be a great a tool to specifically study how different populations may manage limited searching capacity.

We did observe individual differences in the way participants sampling strategies deviate from optimality (see Results section ‘deviations from optimality’). We hypothesise that deviations towards either breadth or depth may partially originate from a participants’ will to reduce uncertainty about either finding a ‘good’ alternative during the sampling phase (one with positive outcome(s)) or estimating correctly the reward during the purchase phase. On one hand, deep sampling enables to better estimate the success probability of the sampled alternatives in order to find the best one but at the risk of not having any good one, as we discussed above in point 3d. This strategy is more reward centred and may be followed by individuals with low risk aversion. On another hand, sampling broadly reduces the risk not to find any good alternative. Such a strategy is more cautious and may be followed by individuals with higher risk aversion profiles. This hypothesis will need to be tested in future studies, by estimating separately participants relation to risk. We also discussed the hypothesis that individual mathematical or probabilistic knowledge may have an effect on the way participants comprehend and behave in the task. The above points were introduced in two paragraphs of the discussion (p.22-23).